# General Articulated Objects Manipulation in Real Images via Part-Aware Diffusion Process

**Zhou Fang**    **Yong-Lu Li**[*]    **Lixin Yang**    **Cewu Lu**[*]

Shanghai Jiao Tong University

{joefang, yonglu_li, siriusyang, lucewu}@sjtu.edu.cn

## Abstract

Articulated object manipulation in real images is a fundamental step in computer and robotic vision tasks. Recently, several image editing methods based on diffusion models have been proposed to manipulate articulated objects according to text prompts. However, these methods often generate weird artifacts or even fail in real images. To this end, we introduce the Part-Aware Diffusion Model to approach the manipulation of articulated objects in real images. First, we develop Abstract 3D Models to represent and manipulate articulated objects efficiently. Then we propose dynamic feature maps to transfer the appearance of objects from input images to edited ones, meanwhile generating the novel-appearing parts reasonably. Extensive experiments are provided to illustrate the advanced manipulation capabilities of our method concerning state-of-the-art editing works. Additionally, we verify our method on 3D articulated object understanding for embodied robot scenarios and the promising results prove that our method supports this task strongly. The project page is at https://mvig-rhos.com/pa_diffusion.

## 1   Introduction

Image editing is a long-standing popular computer vision task. Specifically, manipulating articulated objects has garnered significant attention owing to its application in various fields, such as image augmentation for downstream tasks [44], building goal conditions to train reinforcement learning models for robotic manipulation [42, 55], creating videos with extra supervision information [34], detecting human-object interactions [22, 23, 26], reasoning object affordance [25, 24], *etc*. Thanks to the large-scale training data and immense computing power, diffusion-based [41] generative models have achieved surprising results in the field of image and video generation.

Inspired by these successes, several recent works have adopted diffusion models as the backbone and implemented text-guided object manipulation [20, 16, 8, 51]. We can properly divide these studies into a couple of groups. The first one is to directly edit 2D images by transferring the feature/attention maps from original images to edited ones such as [13, 32, 8]. However, weird artifacts are prone to appear when the objects are rotated and deformed, or novel views appear. Consequently, these methods are restricted to structure-preserving image editing. Another group relies on reconstructing 3D object models. As the most related work to ours, [51] reconstructed 3D object models for manipulation and projected them back to images later. Nevertheless, this approach depends on the quality of reconstructed 3D models heavily. And the reconstruction model has to be fine-tuned when dealing with new categories. Moreover, manipulation has to be done manually which is laborious and impractical to support editing large quantities of images.

To address these problems, we propose the Part-Aware Diffusion Model (PA-Diffusion model) for articulated object manipulation in real images, as illustrated in Fig. 1. Firstly, we introduce the

---

[*]Corresponding authors.

38th Conference on Neural Information Processing Systems (NeurIPS 2024).

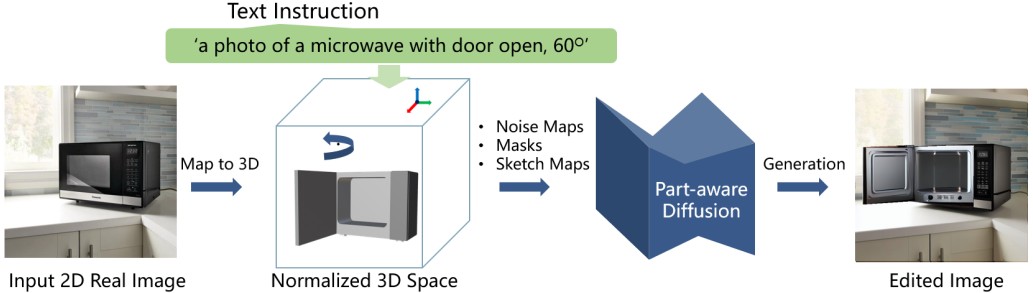

Figure 1: We propose the **Part-Aware Diffusion Model**: Abstract 3D model of the articulated object is constructed referring to the input 2D real image. Various manipulations can be done based on the text instruction or human interaction in the 3D space, the generation model then creates the edited image according to the manipulation.

concept of Abstract 3D Model and build a Primitive Prototype Library to represent articulated objects in the 3D space, so that our method can not only cover many common objects but also handle novel categories without extra training data or fine-tuning processes. Besides, various manipulation can be done efficiently. Second, we proposed dynamic feature maps to assist generation models in accurately transferring object appearances to accurate locations in edited images. As a result, weird artifacts are eliminated, and meanwhile, the novel-appearing parts are generated more reasonably. Finally, owing to the simple manipulation and editing process, the procedure is automatic and the model can strongly support other tasks by editing a large volume of images.

Our main contributions are summarized as follows:

(1) We introduce the concept of the Abstract 3D Model which accurately and robustly represents various articulated object categories with primitive prototypes. Meanwhile, novel categories can also be incorporated quickly. In addition, the articulated objects can be efficiently manipulated with text instructions or human interactions in the 3D space.

(2) We propose dynamic feature maps that let the diffusion model comprehend the object structure. Consequently, the diffusion model can generate novel-appearing parts of objects reasonably and preserve the appearance of the seen parts simultaneously.

(3) We present comprehensive experiments to highlight the advantages of our PA-Diffusion model including comparing with state-of-the-art editing methods both qualitatively and quantitatively, choosing a 3D articulated object understanding experiment to demonstrate how our method supports the tasks in embodied robot scenarios.

## 2 Related Work

### 2.1 Diffusion Model for Image Generation

In recent years, diffusion models [40, 39, 11] have achieved great success in the fields of image/video generation [7, 17], segmentation [5, 50], and many downstream computer vision tasks. To make the generation results controllable, [35] first proposed to extract and incorporate text features into the denoising process. Following this concept, [41, 12, 43, 15, 9] improved the performance of text-guided diffusion models with more effective text embedding methods.

However, as an implicit instruction, text guidance is still not strong enough to finish fine-grained image control such as determining the image layouts, objects' shape and texture, and so on. To make up this gap, [47] provided structural guidance by enhancing the similarity between the features of other conditions and the text guidance. [14, 6] proposed to modify the cross-attention maps and then guide the denoising process. To handle more complex scenarios and achieve more precise control, [52] and [33] proposed adding an extra module to the diffusion model. Then extra condition information can be imported to guide the denoising process.

## 2.2 Diffusion Based Image Editing

Considering the remarkable capability of understanding images, several recent works have also reported editing real and synthetic images with using diffusion models as the backbone. These methods can generally be summarized into two groups: Inversion-Based and Feature-Sharing Based.

The first group is primarily based on adding extra control to the inverted noise maps of images, then re-generating the image such as [20]. However, because the deterministic DDIM sampling process cannot be reversed perfectly, these methods struggle to preserve the appearance of original objects and backgrounds precisely. The second group attempts to maintain the appearance of objects by transferring the feature/attention/activation maps between guidance and generation branches or by adding extra loss items during the denoising process, as seen in [8, 32, 13]. Recent approaches like DragGAN [36] and DragDiffusion [32] propose to utilize a point-to-point dragging scheme, which can achieve refined content dragging. Nonetheless, these approaches often perform poorly on articulated object manipulation in real images, resulting in weird and blurry artifacts in edited images.

2D-3D-2D is another promising way of image editing, the recent work [51] introduced reconstructing 3D models from 2D images and projecting them back after manipulation. However, this approach highly relies on the quality of 3D reconstructed models, and reconstructing 3D models from a single 2D image is still a challenging task.

In contrast to the aforementioned approaches, our method demonstrates advantages when manipulating articulated objects in real images - high fidelity edited images, easy and various manipulations, covering multiple categories, and incorporating novel categories quickly.

# 3 Method

## 3.1 Overview

In this session, we go through the proposed PA-Diffusion model in detail. The overall architecture is demonstrated in Fig. 2. Initially, we reconstruct abstract 3D models for articulated objects with the Primitive Prototype Library. Then various manipulations can be done according to text instructions or human interactions. Next, leveraging DDIM Inversion [45, 31], initial inverted noise maps are created and manipulated following the previous actions. During the generation stage, we introduce dynamic feature maps, including manipulated inverted noise maps and compositional activation maps and images. These ensure that the appearance of seen parts of objects can be preserved accurately and that novel-appearing parts are generated reasonably. Besides, Texture and Style Consistency Score Loss are introduced to alleviate the blurry and style mismatch problems.

## 3.2 Preliminary

Diffusion models aim to convert random Gaussian noise into high-resolution images through a sequential denoising and sampling process [13]. Given the conditioning $y$, we start from the initial Gaussian noise map $z_t$, and then iteratively estimate the reduced noise $\hat{\epsilon}_t$ at each time step $t$:

$$\begin{aligned}
\hat{\epsilon}_t &= \epsilon_\theta(z_t; t, y), \\
z_{t-1} &= update(z_t, \hat{\epsilon}_t, t, t-1, \epsilon_{t-1}),
\end{aligned} \tag{1}$$

The update function could be DDPM [18], DDIM [45], or other sampling methods. Nevertheless, conventional sampling from conditional diffusion models often fails to produce high-quality images that align well with the condition $y$. To enhance the effect of the desired condition, extra class loss guidance is added to the reduced noise during the sampling process such as Classifier or Classifier-free guidance [46, 19].

Classifier guidance is introduced to generate conditional samples from an unconditional model by combining the unconditional score $\epsilon_t$ with a classifier $p(y|z_t)$, where $p(y|z_t)$ is the probability distribution of condition $y$ based on the noise at time step $t$:

$$\hat{\epsilon}_t = \epsilon_\theta(z_t; t, y) + \beta \bigtriangledown_{z_t} p(y|z_t), \tag{2}$$

Classifier-free guidance eliminates the need for a separate classifier by incorporating the class information directly into the generative model as follows:

$$\hat{\epsilon}_t = (1 + \alpha)\epsilon_\theta(z_t; t, y) - \alpha\epsilon_\theta(z_t; t), \tag{3}$$

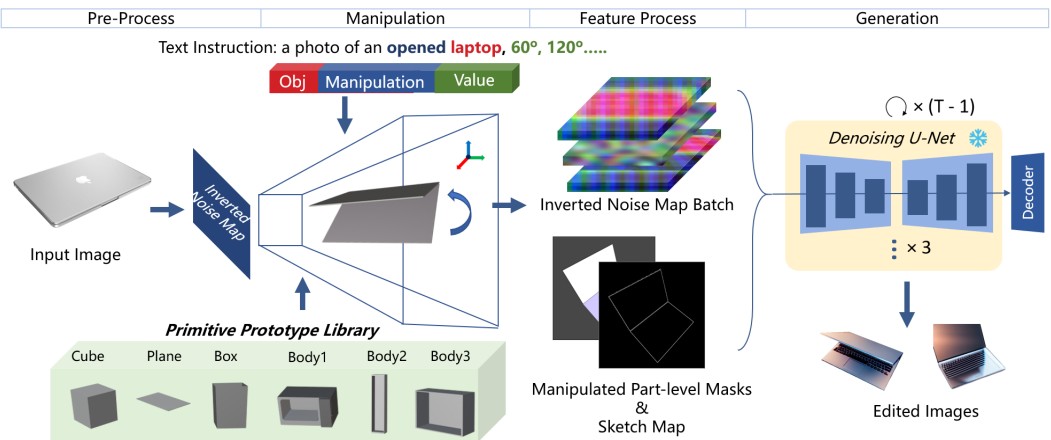

Figure 2: The overall image editing process. (1) In the Pre-Process stage, articulated objects in 2D images are part-level segmented and reconstructed to abstract 3D models. Meanwhile, inverted noise maps of input images are created with DDIM Inversion. (2) In the Manipulation stage, various manipulations can be implemented based on text instructions or human interaction in the 3D space. (3) After manipulation, part-level masks and sketches are rendered and exported. The inverted noise maps are transformed according to these masks. (4) Finally, with the transformed inverted noise maps, sketch maps, and part-level masks, the generation model creates the edited images.

Following these concepts, custom energy functions can also be utilized to guide the denoising process, instead of the probability function. In [13] [30] [54], various energy functions $g$ are incorporated alongside classifier-free guidance to obtain high-fidelity samples as follows:

$$\hat{\epsilon}_t = (1 + \alpha)\epsilon_\theta(z_t; t, y) - \alpha\epsilon_\theta(z_t; t) + \beta \bigtriangledown_{z_t} g(z_t; t, y), \tag{4}$$

Our proposed PA-Diffusion model is built on the diffusion model with classifier-free guidance. Extra energy functions are employed during the image editing process.

### 3.3 Various Manipulations in the 3D Space

As a promising workaround to the methods of dealing with images directly, the 2D-3D-2D pipeline has successfully handled many articulated objects with precise 3D models. Unfortunately, creating 3D models for various categories from a single image remains challenging, particularly for novel categories or instances. In this work, we introduce the concept of Abstract 3D Model that reconstructs accurate 3D models, supports efficient object manipulation, and incorporates novel objects easily.

**Abstract 3D Model.** Unlike previous methods, there is no need for precise 3D models of our method, the conditional information we have to provide to the diffusion model is coarse sketch maps and part-level masks. Therefore, we introduce the use of an abstract 3D model to represent the articulated object. As an abstract 3D model, the object is represented by combining several basic prototypes. As depicted at the bottom of Fig. 2, the laptop can be represented by two planes, storage furnitures and microwaves can be represented by a plane and a box. Primitive Prototype Library, which includes basic 3D prototypes such as planes, cubes, and boxes, can support common articulated object categories involving both rotation and translation joint types.

**Camera alignment.** Next, we compute the camera pose in the 3D space and align the 2D real image view with the 3D space camera view. The pose computation problem is to calculate the intrinsic and extrinsic matrices for the camera that minimize the reprojection error from 3D-2D point correspondences [3]. Thus, in this work, we first employ Large-scale Segmentation Models to obtain the initial part-level segmentation masks of articulated objects $M^{Init}$ and then detect the extreme corner points $A, B, C, D$ ($pts_1$) with simple corner detection functions. These 2D extreme points are aligned with their 3D counterparts $A^{'}, B^{'}, C^{'}, D^{'}$ ($pts_2$, pre-defined in Primitive Prototype Library) as shown in Fig. 3. Finally, based on Perspective n-Points 2D-3D method [49], the camera matrices can be extracted, and 2D-3D views are aligned.

**Manipulation.** By representing objects with primitive prototypes, multiple types of manipulations can be implemented in the 3D space efficiently with the assistance of 3D computer graphics software.

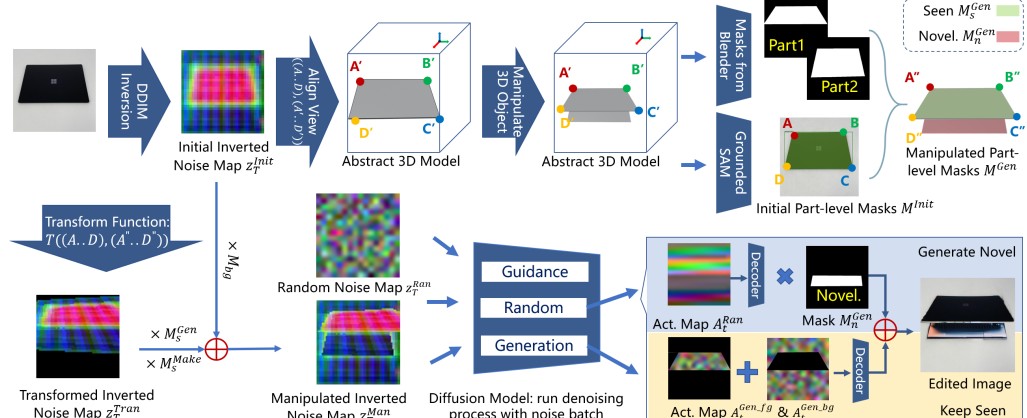

Figure 3: Algorithm pipeline of the PA-Diffusion model. Symbols and procedures in the figure are the same as those in the content.

As shown in Fig. 2, manipulating objects through text instructions is a straightforward approach. For example, the instruction *opening the laptop* 120° is converted to a script, then the manipulation will be done by running the script in 3D software. Our PA-Diffusion model also supports human interaction, which could be even more efficient. Additional manipulation guidance is provided in the Appendix. Contrary to the tedious manipulation experience of previous SOTA works, our proposed PA-Diffusion model offers a more flexible approach to editing articulated objects.

**Structure disentangle.** Some object parts could still be seen after manipulation, and some unseen parts would appear. As shown in the top right part in Fig. 3, the laptop shell can be seen in the input image. After opening the laptop, the shell can still be seen, however, the keyboard is newly revealed. Therefore, to distinguish them and implement part-aware diffusion, we disentangle articulated objects into **seen parts** and **novel-appearing parts**. The appearance of seen parts should be consistent between input and edited images, and the style of novel-appearing parts should be consistent with the objects' overall appearance. In this work, we regard the initial part-level masks $M^{Init}$ as seen. Then after manipulation, we export manipulated part-level masks from the 3D software. And then manipulated seen parts mask $M_s^{Gen}$ and manipulated novel-appearing parts mask $M_n^{Gen}$ can be calculated.

### 3.4 Dynamic Feature Maps

To maintain the object's appearance including color and texture, previous editing methods introduced a **guidance branch** to invert and re-generate the input image, and a **generation branch** to create the edited image, the attention/feature/activation maps are transferred from the guidance to the generation branch directly [8]. However, when changing the object location or shape during manipulation, directly sharing these maps would transfer the feature from the input image to undesired locations in the edited image. Furthermore, these methods cannot reasonably generate novel views or novel-appearing parts.

To overcome these problems, we propose dynamic feature maps including manipulated inverted noise maps and compositional activation maps and images. To keep appearance accurate, manipulated inverted noise maps transfer the feature of seen parts in input images to the manipulated seen parts in edited images. Simultaneously, to make novel-appearing parts reasonably, compositional images let the diffusion model create these parts from random noise. The following content describes how to manipulate the noise maps, how to construct compositional maps, and how they work. For clarity, the process is presented in Fig. 3.

**Manipulated inverted noise map.** As shown at the top of Fig. 3, we firstly reverse the input image to the initial inverted noise map $z_T^{Init}$ with DDIM inversion. After 3D manipulation, we calculate the transform function $T$ based on the initial $M^{Init}$ and manipulated seen part-level masks $M_s^{Gen}$ and then compute the transformed inverted noise map $z_T^{Tran}$ with $T$. The manipulated inverted noise map is created as the following equation:

$$z_T^{Man} = z_T^{Tran} \times M_s^{Gen} + z_T^{Tran} \times M_s^{Make} + z_T^{Init} \times M_{bg}, \tag{5}$$

where $M_s^{Make}$ is the makeup mask generated by $XOR(M^{Init}, M^{Init} \cap M_s^{Gen})$. $M_{bg}$ is the background mask created by $1 - M_s^{Gen}$. In addition, we also create a random noise map $z_T^{Ran}$. The three noise maps $z_T^{Init}$, $z_T^{Ran}$, and $z_T^{Man}$ will be sent to the denoising UNet in the next step.

**Compositional activation map and image.** As shown at the bottom of Fig.3, the diffusion model runs a sequential denoising process with the three noise maps as a batch. The activation map $A_t^{Gen\_fg}$ and $A_t^{Gen\_bg}$ for seen parts and background are generated separately with the supervision of the guidance branch, and then merged to build $A_t^{Gen}$ according to the seen part mask $M_s^{Gen}$. After the decoder, the initial edited images $I^{Gen}$ are created. Owing to the above steps, we transfer the feature of seen parts in input images to the accurate location in edited images. The other contents and the background can also be preserved, as the highlighted yellow part in Fig. 3. Besides, as the highlighted blue part in Fig. 3, an extra image $I^{Ran}$ is synthesized with random noise map $z_T^{Ran}$, the novel-appearing parts are cropped and pasted to edited images $I^{Gen}$ as follows, which makes these parts more reasonable and consistent with the original inputs.

$$I^{Gen} = I^{Ran} \times M_n^{Gen} + I^{Gen} \times \left(1 - M_n^{Gen}\right), \tag{6}$$

### 3.5 Score Function

**Texture Consistency Score Loss.** However, simply manipulating the inverted noise map will lead to a serious blurry problem. This is due to the denoising process includes several convolution steps. As the initial inverted noise map $z_T^{Init}$ is not rotation invariant, manipulating $z_T^{Init}$ will disturb the original distribution and make the denoising process fail. To alleviate this limitation, we construct Texture Consistency Score Loss (TCSL) [32] as an extra supervision that lets the specific region in the generation branch match with the one in the guidance branch,

$$\begin{aligned} Loss_t^{fg} &= \frac{1}{cos(F_t^{Gui}[M^{Init}], F_t^{Gen}[M_s^{Gen}])} \\ Loss_t^{bg} &= \frac{1}{cos(F_t^{Gui}[1 - M^{Init}], F_t^{Gen}[1 - M_s^{Gen}])}, \end{aligned} \tag{7}$$

where $F_t^{Gui}$ and $F_t^{Gen}$ are feature maps of the guidance and generation branch. We add these items as extra losses in classifier guidance in each denoising iteration step to calibrate the appearance of objects and background.

**Style Consistency Score Loss.** For novel-appearing parts, the diffusion model is prone to randomly select a style to generate them with text guidance or sketch maps. As a result, the texture and style could be different from the objects in input images. Therefore, we introduce Style Consistency Score Loss (SCSL) to calibrate the style of seen parts and the novel-appearing parts.

Different from Texture Consistency Score Loss, there is no need to match every pixel in input images and edited images. Thus we calculate L1 loss between the feature maps $F_t^{Ran}$ of the random branch and $F_t^{Gui}$ [13]. The loss function is as follows:

$$Loss_s = \left| sum(F_t^{Gui}[M^{Init}]) - sum(F_t^{Ran}[M_n^{Gen}]) \right|_1, \tag{8}$$

Briefly, the reduced noise $\hat{\epsilon}_t^{Gen\_fg}$, $\hat{\epsilon}_t^{Gen\_bg}$, and $\hat{\epsilon}_t^{Ran}$ in each denoising iteration can be presented as follows, where $\gamma_1$, $\gamma_2$, and $\gamma_3$ are the hyper-parameters.

$$\begin{aligned} \hat{\epsilon}_t^{Gen\_fg} &= \hat{\epsilon}_t^{Gen\_fg} + \gamma_1 \nabla_{z_t} Loss_t^{fg} \times M_s^{Gen} \\ \hat{\epsilon}_t^{Gen\_bg} &= \hat{\epsilon}_t^{Gen\_bg} + \gamma_2 \nabla_{z_t} Loss_t^{bg} \times (1 - M_s^{Gen}) \\ \hat{\epsilon}_t^{Ran} &= \hat{\epsilon}_t^{Ran} + \gamma_3 \nabla_{z_t} Loss_s \times M_n^{Gen} \end{aligned} \tag{9}$$

## 4 Experiment

In this section, we provide two kinds of experiments to prove the advantages of our proposed PA-Diffusion model. First, various image editing tasks are conducted to showcase the model's image editing capabilities. To highlight the superiority of our model compared with state-of-the-art methods, we collect a testbench and evaluate all the methods both qualitatively and quantitatively. Second, we create a synthetic training set to support the challenging 3D articulated object understanding task in the robotic scenarios.

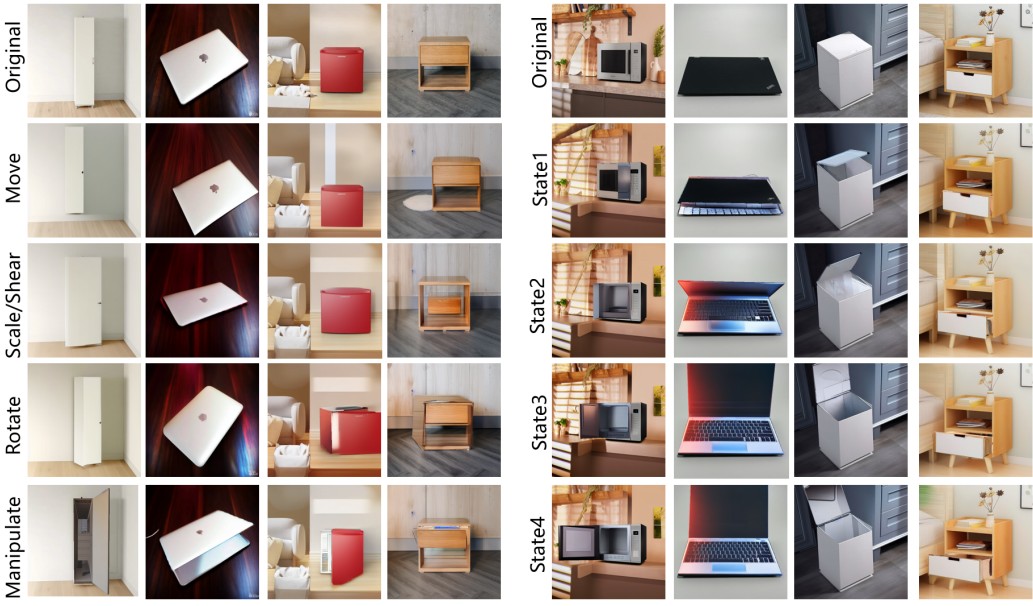

Figure 4: Results of basic manipulations: move, scale/shear. rotate, and manipulate. Blank regions caused by the manipulation are in-painted automatically. The novel views and novel-appearing parts match with the style of the seen parts (left). Articulated objects are opened from the initial close state with 4 steps. The appearance of the increasing novel-appearing parts keeps being consistent throughout the whole process (right).

## 4.1 Implementation

In this work, we select Grounded Segment Anything [21, 29] to obtain the initial part-level object segmentation masks. T2I Adapter [33] is chosen as the conditional generation model, and the condition we used is the sketch map. The fundamental diffusion model is Stable Diffusion V1-5. All experiments run on a single NVIDIA A100 GPU. Notably, **NO** models need to be trained or fine-tuned in the image editing process.

Primitive Prototype Library is built within Blender [10]. 3D planes, cubes, boxes, and other 3D primitive shapes are created and combined to represent different objects. In this work, 5 primitive shapes are collected and combined to represent 6 categories of articulated objects. The ease of creating prototypes allows for the quick incorporation of novel categories or instances. Various manipulations can be implemented in Blender efficiently.

## 4.2 Results

Fig. 4 demonstrates the editing results of some basic manipulations and a sequential manipulation process. As shown in the left part, our PA-Diffusion model naturally moves, scales/shears, rotates and opens articulated objects with rotation or translation joint types. The edited objects blend seamlessly with other contents and backgrounds in the original images. When we move or rotate the objects, the blank regions in the background are in-painted semantically according to the surroundings. Moreover, in-painting and editing are completed in a single denoising process by the PA-Diffusion model, no extra in-paint model or process is required. Last but not least, novel-appearing parts of objects are generated reasonably, and the style matches with the object. For example, the storage furniture is empty after opening, and the keyboard of the laptop is designed logically.

The right part of Fig. 4 presents a complete operation process of opening articulated objects, from the initial closed state to the open state with 4 steps progressively. The appearance of objects' seen parts is transferred from the original input to different states accurately. The point that needs to be mentioned is that along with the operation progress, more novel-appearing parts of objects appear, our proposed PA-Diffusion model keeps the style and texture of these novel-appearing parts being consistent throughout the process. This capability potentially allows our method to generate a

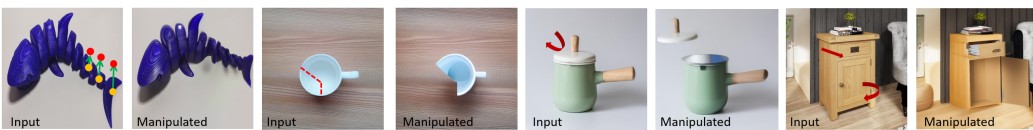

Figure 5: Manipulate non-rigid objects, non-uniform shapes, and objects with weird or multiple joint types.

complete manipulation video from a single input image, maintaining consistent object appearance and style even as novel views or parts increase.

Generally, articulated objects' parts are rigid and connected with one of the typical joint types - rotation and translation. However, we are surprised to notice that our PA-Diffusion model can also handle non-rigid objects with non-uniform shapes, weird joints, and manipulations. As illustrated in Fig. 5, we first select toys as examples of non-rigid with non-uniform shapes, the tail of the shark is moved up together with other close parts similar to deformable objects. Then, we broke the cup in a real image. Third, the kitchen pot and storage furniture are opened to illustrate the case of weird and multiple joint types within one object. No matter whether the shape of object parts has changed after manipulation or joint types are unconventional, the PA-Diffusion model can edit the objects successfully. Meanwhile, the background is preserved or inpainted well.

### 4.3 Ablation Study

As mentioned in Section 3, TCSL is added to the denoising loss function to release the serious blurry problem. As shown in the left of Fig. 6, we move the storage furniture, the microwave, and the drawer to different directions. It can be seen that without TCSL, the objects are prone to be blurry. More seriously, the edited image could be destroyed, as in the example of the drawer. On the other hand, with TCSL, the texture of objects can be transferred to the desired location, meanwhile, the blank region caused by the movement is in-painted well. SCSL is another loss to keep the style consistent between seen parts and novel-appearing parts. Its effectiveness is shown in the left bottom of Fig. 6 (with SCSL). We notice that when opening the storage furniture, the style of the novel-appeared inner door and body part is more likely to be consistent with the outer body part with SCSL, which makes the edited image natural. More quantitative ablation studies about the two score losses are provided in the Appendix.

### 4.4 Comparison

To further evaluate our PA-Diffusion model, we compare it with four state-of-the-art image editing approaches that are based on diffusion models: Imagic, DragDiffusion, MasaCtrl (with T2I Adapter), and Image Sculpting. In this experiment, we require these models to manipulate different categories of articulated objects, including both rotation and translation joint types. The results are shown in the right part of Fig. 6. It is hard for Imagic to finish the tasks as articulated objects cannot be opened at all or the wrong part is manipulated. This is because the text instruction is too weak and the fundamental generation model cannot understand the structure of objects. Similarly, DragDiffusion cannot finish the tasks even though human interaction is applied.

MasaCtrl performs better than Imagic and DragDiffusion. The manipulation can be finished, while the edited images are prone to be either unrealistic or unreasonable. Take the laptop as an example (second column in the right part of Fig. 6), the object has been opened, while the region highlighted with the red bounding box in the edited image remains unchanged, which does not make sense. This issue is prevalent across other categories as well. The reason is that MasaCtrl simply shares the whole feature/attention maps between the input and edited image, features of seen parts cannot be transferred to the desired new location when the objects move or the shape changes. Finally, Image Sculpting works well on the storage furniture. However it is likely to fail to reconstruct precise 3D models of the laptop, trashcan, and drawer, consequently, the edited images are undesirable.

In contrast, our PA-Diffusion model consistently produces high-fidelity and reasonable edits. The appearance of seen parts is kept accurate no matter whether objects move or the shapes of parts have changed. The novel parts are reasonable and semantically consistent with the original.

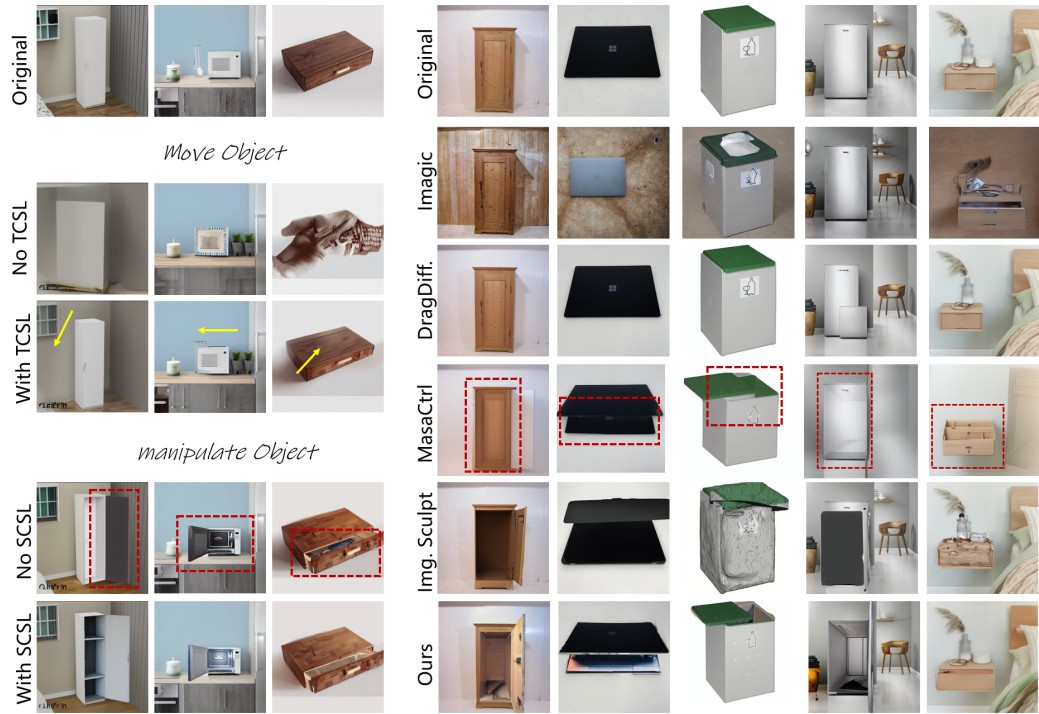

Figure 6: TCSL and SCSL are employed to release the blurry and style mismatch problem. The model is required to move and open the object (left). Comparison of Imagic, DragDiffusion, MasaCtrl (with T2I adapter), Image Sculpting, and our PA-Diffusion model. The target state is *'a photo of an opened object'* (right).

## 4.5   Quantitative Evaluation

To quantitatively evaluate our method, we built an articulated object manipulation testbench. The testbench comprises 6 object categories including storage furniture, laptop, trashcan, microwave, drawer, and refrigerator, which covers both rotation and translation joint types. In total, 660 real images are collected from the website. Considering articulated objects are typically rigid with uniform shapes, this testbench can represent the characteristics of common articulated object categories.

The comparison methods we select are Imagic and MasaCtrl (with T2I Adapter). DragDiffusion is excluded here, as it simply reconstructs the original image and cannot complete the manipulation tasks. Due to the long manual processing time and frequent failures in generating 3D models, Image Sculpting is also excluded here. To assess the realism of the edited images, the evaluation metric used is the Frechet Inception Distance (FID) score. The quantitative evaluation results are summarized in Tab. 1.

Since Imagic relies solely on text instructions, the edited images often do not align well with the original inputs, resulting in poor scores. Due to previously discussed reasons, the edited images of MasaCtrl are confusing and lack coherence. Sequentially, the FID score is not satisfying. In comparison, the PA-Diffusion model outperforms other methods with over 40.1% improvement.

## 4.6   Articulated Object Understanding

In this session, we demonstrate how our proposed method supports the task of 3D articulated object understanding. As one of the fundamental steps to understanding 3D articulated objects, estimating the axes and surface normal is still challenging because of the lack of data. To release the data limitation, we create a synthetic dataset with the PA-Diffusion model. The dataset includes 660 sequential samples, each sample includes the sequence of opening objects from the close state with 4 steps. [38] introduced a 3-step training process to develop the object understanding model including

| Category | Imagic ↓ | MasaCtrl ↓ | **Ours ↓** |
|---|---|---|---|
| Storage. | 5.85 | 2.27 | 0.81 |
| Laptop | 9.48 | 1.17 | 1.94 |
| Microwave | 1.34 | 3.15 | 0.87 |
| Trashcan | 5.62 | 1.59 | 0.96 |
| Refrigerator | 7.75 | 1.76 | 1.38 |
| Drawer | 30.7 | 0.98 | 0.60 |
| Avg. | 10.1 | 1.82 | **1.09** |

Table 1: FID Score of edited images with Imagic, MasaCtrl (with T2I adapter) and ours.

| Category | bbox ↑ | bbox+axis ↑ | normal < 30° ↑ |
|---|---|---|---|
| Storage. | 67.5/65.0 | 65.0/65.0 | 59.9/61.5 |
| Laptop | 87.5/90.0 | 64.2/75.2 | 22.8/35.7 |
| Microwave | 80.0/85.0 | 80.0/85.0 | 72.4/70.7 |
| Trashcan | 95.0/92.5 | 74.4/84.9 | 40.5/49.2 |
| Refrigerator | 70.0/80.0 | 70.0/80.0 | 63.2/80.0 |
| Drawer | 70.0/82.5 | 70.0/78.8 | 67.5/82.5 |
| Avg. | 78.3/**82.5** | 70.6/**78.2** | 54.1/**63.2** |

Table 2: Prediction accuracy of the model developed with half (left) and full (right) training set separately.

| Dataset | AUROC ↑ | bbox ↑ | bbox+axis(rot) ↑ | bbox+axis(rot) normal ↑ | bbox ↑ | bbox+axis(tran) ↑ | bbox+axis(tran) normal ↑ |
|---|---|---|---|---|---|---|---|
| InternetVideo | 74.0 | 62.1 | 28.5 | 16.5 | 32.0 | 26.2 | 14.3 |
| **Mixed** | **74.7** | **66.1** | **29.2** | **18.3** | **38.1** | **30.0** | **19.3** |

Table 3: Mix the edited images with the training set of the InternetVideo dataset, then evaluate the fine-tuned model on the testing set of the InternetVideo dataset.

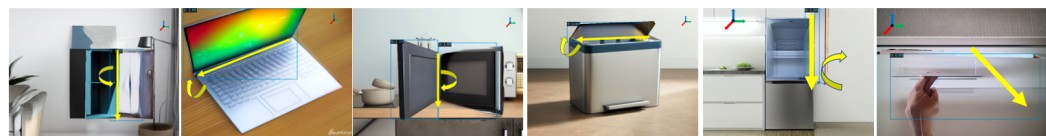

Figure 7: Detection results on the testing samples, including rotation and translation joint types.

BBox detection, axis prediction, and plane normal estimation. Following this schedule, we evaluate the feasibility of edited images by two kinds of experiments.

First, the generated sequential samples are divided into training/testing sets (612/48). Specifically, we follow the 3-step to develop the model with half and full training samples separately, and then evaluate the model with three matrices, BBox IoU, Axes EA-score, and surface normal error smaller than 30° [38]. Fig. 7 demonstrates the prediction results, the model can understand the structures of articulated objects after training with edited images, including moving plane, joint types, axis and surface normal. Quantitative evaluation results in Tab. 2 indicate that the prediction accuracy improves with more training samples, illustrating that the edited images are comparable to the real ones.

Second, to further evaluate the edited images, we merge them with the original training set of Internet Video dataset [38] and fine-tune the pre-trained model. The fine-tuned model is evaluated on the testing set (6,231 real images) of the InternetVideo Dataset. $Baseline$ refers to the model trained on the InternetVideo dataset only. Here, we select to use the same evaluation matrices as [38], surface normal accuracy is multiplied with the accuracy of BBox and axis, other evaluation metrics are the same as above. The evaluation results are summarized in Tab. 3. Compared with the baseline, the overall performance has been improved by enlarging the training set with edited images. The above two experiments illustrate how our PA-Diffusion model can benefit robotic vision tasks.

## 5 Limitations

Even though our method can handle common articulated objects, there are still some limitations. First, as edited images are generated from inverted noise maps, the quality of the original input images significantly affects the editing outcomes. Blurry or low-resolution inputs will degrade the edited images. Second, when the object undergoes substantial deformation or movement, this editing method is likely to fail. Besides, manipulating deformable objects and fluids remains challenging with this approach. Further explanation is provided in the Appendix.

## 6 Conclusion

This work introduces the PA-Diffusion model, a novel articulated object manipulation method that covers common object categories and supports various manipulations. Both the qualitative and quantitative experiments have proven the feasibility and effectiveness of our method. Besides, the 3D articulated object understanding experiment illustrates that the PA-Diffusion model has positive impacts on helping build robots that interact with the real world smartly.

## Acknowledgments and Disclosure of Funding

This work is supported by the National Natural Science Foundation of China under Grants 62306175, the National Key R&D Program of China (No.2021ZD0110704), CCF-Tencent Rhino-Bird Open Research Fund, the National Key Research, Development Project of China (No.2022ZD0160102), the National Key Research and Development Project of China (No.2021ZD0110704), Shanghai Artificial Intelligence Laboratory, XPLORER PRIZE grants, STCSM 2023 Pujiang X Program Project (No.23511103104), and STCSM 2024 Qimingxing-Yangfan Fund (No.24YF2722000).

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

# Appendix

In this appendix session, we first go through the pipeline of the PA-Diffusion model, and then provide more experiment results and detailed explanations, the arrangement is as follows:

## A   Additional algorithm pipeline of the PA-Diffusion model

To facilitate the understanding of our proposed PA-Diffusion model, we present the entire algorithm pipeline in Algorithm 1. $eq.5, eq.7,$ $and$ $eq.8$ refer to the equations in the main paper.

---

**Algorithm 1** PA-Diffusion Model

---

**Require:** Manipulate the articulated objects in RGB images
**Input:** RGB image $x$, Primitive Prototype Library
**Output:** Edited RGB image $x_{edit}$
**Pre-Process:**
  1: Generate initial inverted noise map $z_T^{Init}$ with $DDIM\,Invertion$
  2: Generate initial part-level masks $M^{Init}$ with $Grounded\,SAM$
  3: Create 3D Abstract model and calibrate 2D image - 3D camera view
**Manipulation:**
  4: Manipulate articulated objects with text instructions or human interaction in the 3D space
  5: Calculate manipulated part-level masks $M_s^{Gen}$ of seen part and $M_n^{Gen}$ of novel-appearing part
**Feature Process:**
  6: Calculated manipulated inverted noise map $z_T^{Man}$ as $eq.5$.
**Generation:**
  7: Send initial $z_T^{Init}$, random $z_T^{Ran}$, and manipulated $z_T^{Man}$ inverted noise map to diffusion model
  8: **for** $t = T, ..., 1$ **do**
        Add extra loss items TCSL $Loss_t$ and SCSL $Loss_s$ as $eq.7$ and $eq.8$
        Create activation map $A_t^{Gen}$
      **end**
  9: **Output:** Edited image $x_{edit} = Decoder(z_0)$

---

**Dynamic Feature Maps:** To make it clear, we simplify the $64 \times 64 \times 4$ inverted noise maps and compositional images to pure color block maps, as shown in Fig. 8. Simple actions such as moving, scaling, and shearing can be implemented with grid sample[4] directly. However, for complex actions, perspective transform [48] is required. As the following equations show, we calculate the perspective transform function $Transform$ based on $pts_1$ and $pts_3$, where $pts_1$ and $pts_3$ are corner points of the input image masks $A, B, C, D$ ($pts_1$) and corner points of manipulated masks $A^", B^", C^", D^"$ ($pts_3$) as introduced in the main paper. The corner points are automatically detected with a simple corner detection function. And then we can get the transformed inverted noise map $z_T^{Tran}$ by transforming the initial inverted noise map $z_T^{Init}$ as following equations. Finally, the manipulated inverted noise map is calculated by adding $z_T^{Tran}$ and $z_T^{Init}$.

$$T = Transform(pts_1, pts_3),$$
$$z_T^{Tran} = T(z_T^{Init}), \tag{10}$$

On the other hand, the edited images are generated by adding $I^{Ran}$ generated with the random noise map and $I^{Gen}$ generated with the manipulated inverted noise map, as shown in the right part of Fig. 8.

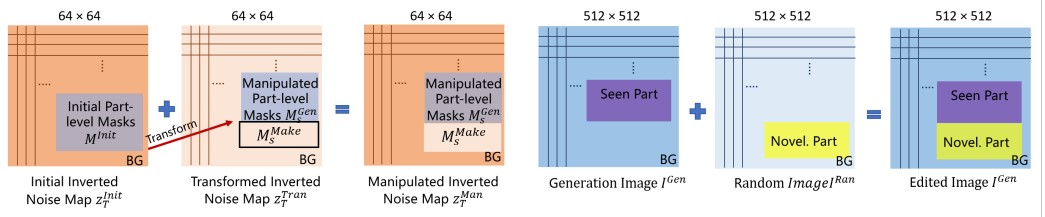

Figure 8: Manipulating noise maps and constructing compositional images. The manipulation is implemented with grid sample or perspective projection. The final edited image is constructed by merging two activation maps and images.

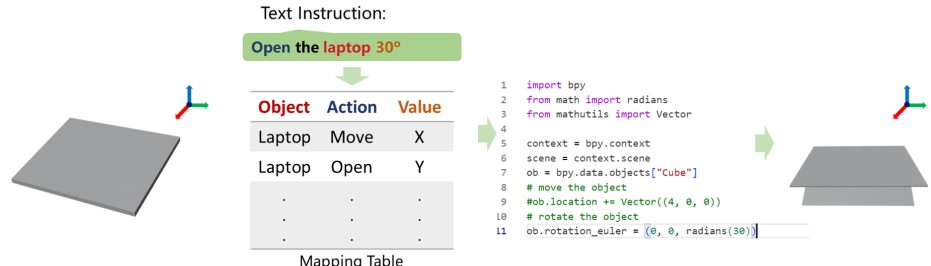

Figure 9: Example of manipulating 3D objects in Blender with text instructions.

**Articulated Object Manipulation in 3D space:** As noted in the main paper, different manipulations can be done with text instructions or human interaction. For the text-based method, considering the objects can be manipulated with Python scripts in Blender, we first need to construct a table mapping the text instructions to actions. Then these actions can be implemented by running Python scripts. As shown in Fig. 9, we require the laptop to open 30 degree, this text instruction is converted to Python script. Finally, running this script can finish the open action. For the second type, users can directly manipulate any parts of articulated objects in Blender, which could be more flexible and convenient.

## B  Additional Manipulation Results

In Fig. 10 and Fig. 11, more articulated object manipulation results synthesized by our proposed PA-Diffusion model are demonstrated. Novel categories including door, toilet, and book are experimented here. For various categories, joint types, and backgrounds, our proposed method can manipulate the objects and preserve other contents in the input images simultaneously.

## C  Additional Ablation Study

To analyze and explain our proposed PA-Diffusion model in detail, we provide more ablation studies in this session.

**Additional Loss Items:**  In the main paper, we qualitatively demonstrate the effect of TCSL and SCSL in the experiment part. Here, we evaluate them quantitatively. Following the main paper, the evaluation metric is the FID score. The results are summarized in Tab. 4. It is obvious that without TCSL, the performance degrades significantly since the feature cannot be transferred to the edited images correctly leading to inconsistent appearance compared to the input images. On the other hand, SCSL only aims to adjust the style of novel-appearing parts, which does not determine the score. When including the two losses, the edited images are close to the input real images in the aspects of color, texture, and style.

**Primitive Prototype Library:**  In the main paper, we collect 5 primitive prototypes to represent 6 different kinds of articulated objects in the testbench. Since our method does not require precise 3D CAD models of objects, Primitive Prototype Library can cover a wide range of articulated objects with a small number of primitive prototypes. As shown in Fig. 10 and Fig. 11, books can be represented as laptops, doors can be represented with simple planes, and toilets can be represented with a plane and boxes. No extra prototypes are required when creating abstract 3D models for

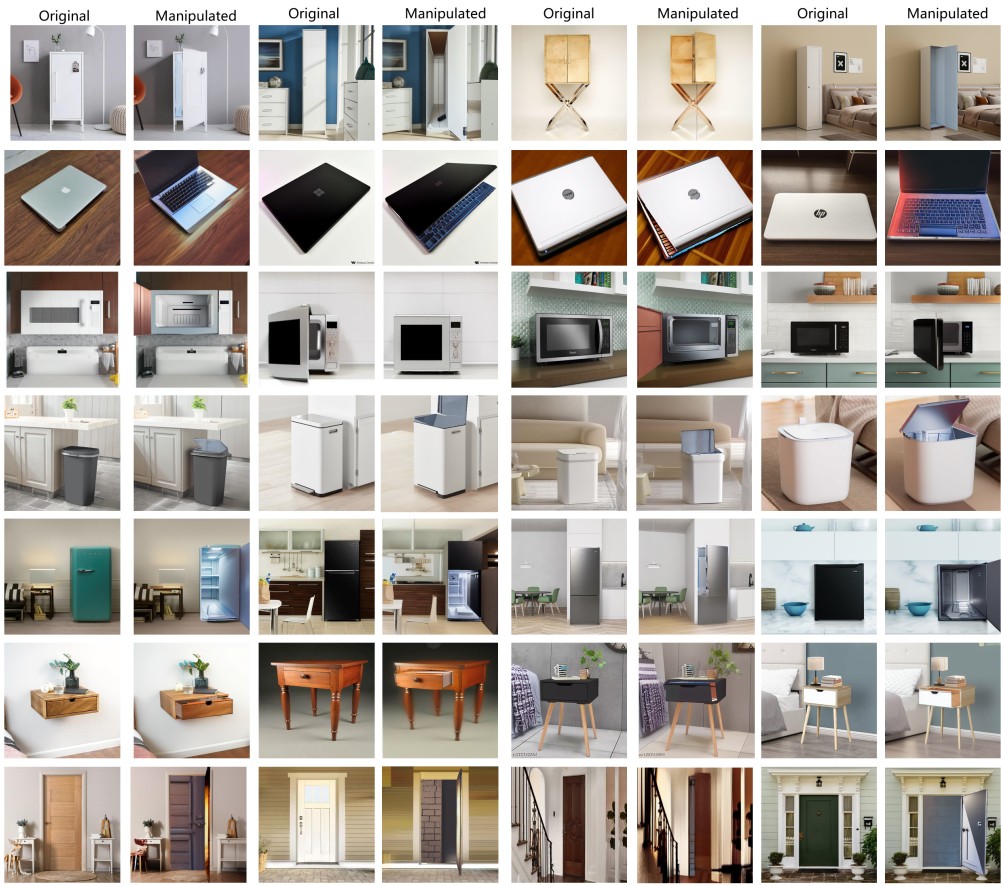

Figure 10: Additional demonstration of the images edited with our proposed PA-Diffusion model, including storage furniture, laptop, microwave, trashcan, door, drawer, and refrigerator.

| Category | None ↓ | only SCSL ↓ | only TCSL ↓ | All Losses ↓ |
|---|---|---|---|---|
| Storage. | 1.48 | 1.54 | 0.99 | 0.81 |
| Laptop | 2.30 | 1.92 | 1.88 | 1.94 |
| Microwave | 1.52 | 1.71 | 0.71 | 0.87 |
| Trashcan | 1.39 | 1.65 | 0.70 | 0.95 |
| Refrigerator | 1.99 | 1.81 | 1.79 | 1.38 |
| Drawer | 1.63 | 1.25 | 0.51 | 0.60 |
| Avg. | 1.72 | 1.65 | 1.10 | **1.09** |

Table 4: FID score of edited images with different additional losses: no SCSL and TCSL, with SCSL only, with TCSL only, and with all losses. The performance improves $36.6\%$ with the assistance of the two losses.

novel categories. Besides, the edited images are still high-fidelity and high-quality. Furthermore, the Primitive Prototype Library is easy to expand, more primitive prototypes can be created or collected rapidly when dealing with novel articulated objects.

**2D-3D Models analysis:** Besides the advantage of convenience, we also compare the quality of reconstructed 3D models with state-of-the-art 2D-3D methods and abstract 3D models.

Following the method introduced in Image Sculpting [51], we use ClipDrop to remove the background of input images, and then reconstruct 3D models with Zero123 [28]. The tested images are the same as those in the main paper. The reconstructed 3D object mesh models are shown in Fig. 12 including the front view, side view, and the edited images after manipulation. We notice that the result of storage furniture is acceptable, the shape closely matches the original, and the texture is stored in UV

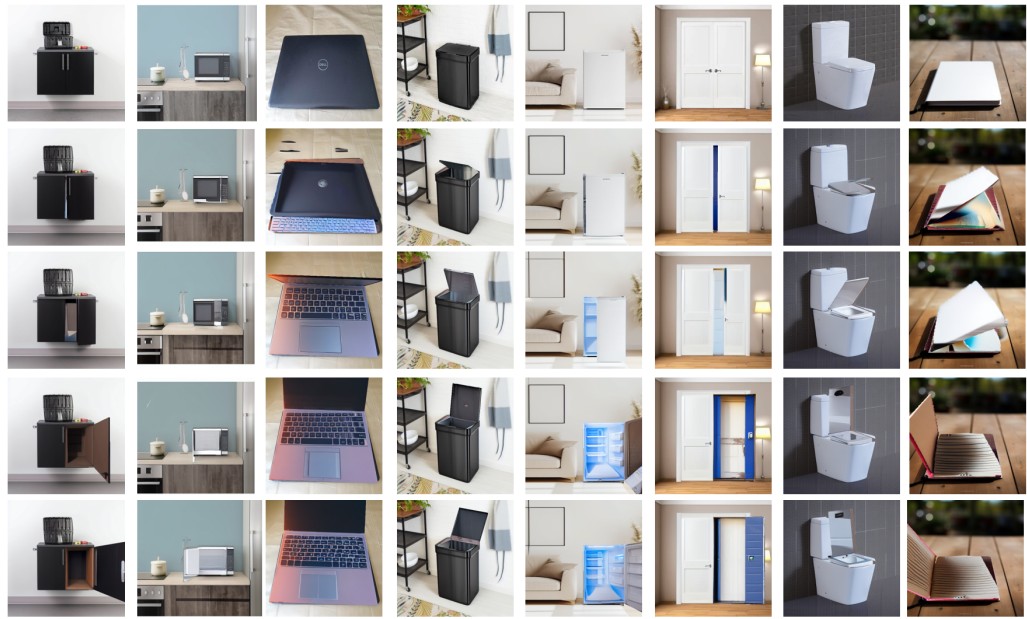

Figure 11: Additional demonstration of editing images with manipulation process based on our proposed PA-Diffusion model. The last three columns are novel articulated object categories.

maps correctly. However, for other categories, the reconstructed models are poor which is the main reason for low-quality edited images.

In the main paper, we mentioned that manipulating the reconstructed 3D models created by 2D-3D methods is tedious and inaccurate. As shown in Fig. 12, the reconstructed models are not part-level (only one mesh object), tremendous human effort is required to cut the mesh into parts before manipulation. For example, when trying to split the door from the body of storage furniture, we need to cut the furniture into several parts first (top, bottom, four sides of the body), and then merge others except the door. This process, even with advanced 3D graphics software, is complex and labor-intensive. As a result, summarizing reconstruction, manipulation, and generation time, Image Sculpting [51] requires over 10 mins to manipulate one image, making it unsuitable for large-scale image editing tasks.

In summary, using abstract 3D models to present articulated objects offers several advantages: (1) State-of-the-art 2D-3D methods are still not robust enough to create precise 3D models such as laptops and trashcans. In comparison, it is easy to achieve abstract 3D models with primitive prototypes, as shown in Fig. 12. (2) Manipulating primitive prototypes in the 3D space is easier and more accurate than manipulating a single 3D object mesh. (3) Novel categories and instances can be handled with our method efficiently. (4) Our method is brief and thus can support various downstream tasks.

## D   3D Articulated Objects Understanding

**Overview.**   In the main paper, we introduce the 3D articulated object understanding experiment and demonstrate how our proposed PA-Diffusion model supports other research fields. Here, we discuss the experiment setup and implementation in more detail.

**Annotation Generation.**   In previous works [38, 37, 27], human labeling is required for annotating the bounding boxes, rotation/translation axes, and surface normal, which is labor-intensive and inaccurate. On the contrary, by representing objects with abstract 3D models, our method can achieve these annotations automatically.

As shown in Fig. 13, $part1$ and $part2$ are the masks of articulated objects' parts that are exported from Grounded SAM or Blender software. The corner points can be calculated with [1, 2] automatically. Then, the bounding box of each object part can be calculated with these corner points. The rotation and translation axis annotations are represented as $[x_0, y_0, x_1, y_1]$, where $x_0, y_0, x_1, y_1$ are the coordinates

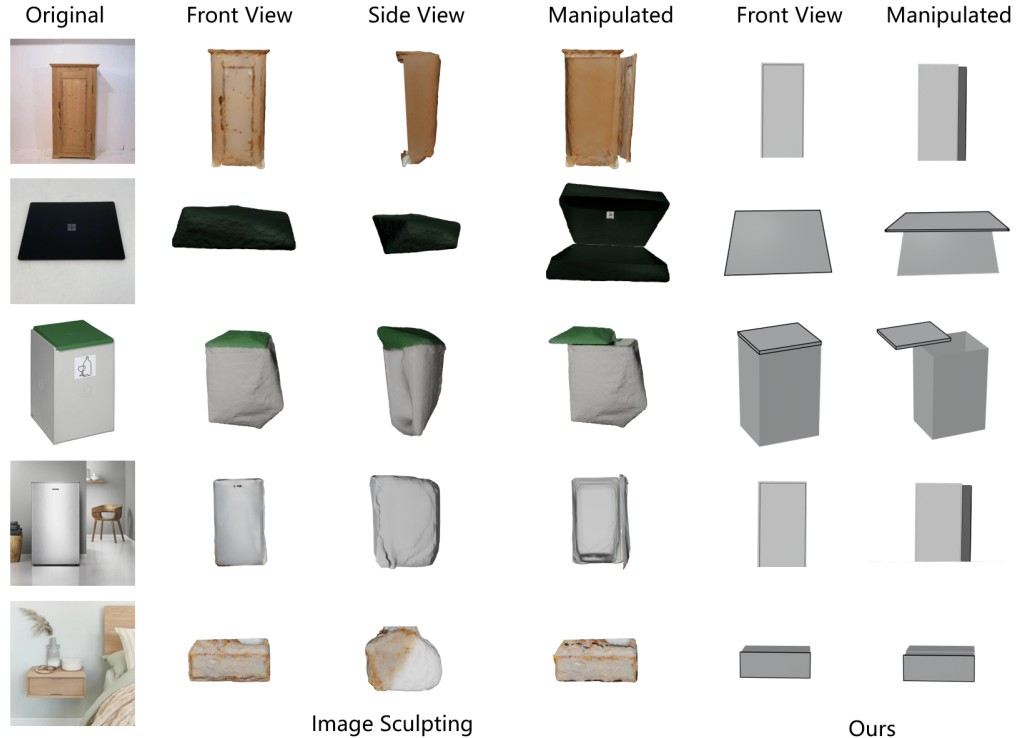

| Original | Front View | Side View | Manipulated | Front View | Manipulated |
|----------|------------|-----------|-------------|------------|-------------|

Image Sculpting           Ours

Figure 12: Reconstructed 3D object models with Image Sculpting, and our abstract 3D models created with Primitive Prototype Library.

of corner points in part-level masks, different object category uses different corner points. For the 3D surface normal annotations, as shown in Fig. 14, we first export the object's part world transform matrix $M_{obj}$ and camera world transform matrix $M_{camera}$. Then the two matrices are normalized and calibrated to create the aligned transform matrix $M_{aligned}$. Consequently, we simply select the normal of one plane to represent the orientation of the object part, and the surface normal $V_{surf}$ is equal to the multiplication of the aligned matrix and local plane vector $V_{plane}$:

$$M_{aligned} = Align(Norm(M_{camera}), Norm(M_{obj})),$$
$$V_{surf} = M_{aligned} \times V_{plane}. \tag{11}$$

**Evaluation Matrix.** For quantitative evaluation, we follow [38] to calculate the average precision of the bounding box, axis, and surface normal. The bounding box is the traditional horizontal type, the threshold of IoU is set as 0.5. The predicted axes are measured with EA score as [53]. To demonstrate the results clearly, we calculate the surface normal error and measure the accuracy that the error is smaller than the threshold $30°$.

## E  Limitations and Future Research

The limitations of our proposed PA-Diffusion models have been discussed in the main paper. Due to the inaccuracy of DDIM inversion, the inverted noise map might be poor if the input image quality is low. Unfortunately, the poor noise map will lead to mismatch error accumulation and propagation during the iterative denoising process. As in the left of Fig. 15, when the original input image is of low resolution (it is normalized to $512 \times 512$ before manipulation), the PA-Diffusion model cannot re-generate the original image with the inverted noise map. Simple actions like moving and scaling also cannot be completed.

Manipulating the initial inverted noise maps is critical to preserve the appearance of seen parts. However, as discussed in the main paper, this step disturbs the original data distribution. The problem will be too serious to be fixed when the object shape deformation is large. As shown in the right of

Fig. 15, when reshaping the laptop to a non-uniform diamond shape, the object's appearance cannot be preserved.

Considering this situation, one promising solution is to add stronger and more precise supervision loss in each denoising step. This is beyond the scope of this work, we plan to implement this later.

In the future, more categories of articulated objects will be covered and the edited image dataset will be expanded to millions-scale for supporting various computer vision and robotic manipulation tasks. Next, we will extend this method to handle deformable objects and fluids.

## F   Societal impacts and potential risks

The articulated object manipulation method presented in this work has profound positive societal implications. This method can serve as a fundamental tool to benefit other computer vision or robot vision tasks. Consequently, the artificial intelligent algorithm can understand and interact with the real world better. Humans will have stronger AI assistants including smart offices, intelligent home or medical robots, and so on.

All the models and data used in this work are collected from the Website. No personal information is used. The code and data created in this work have a low risk of misuse.

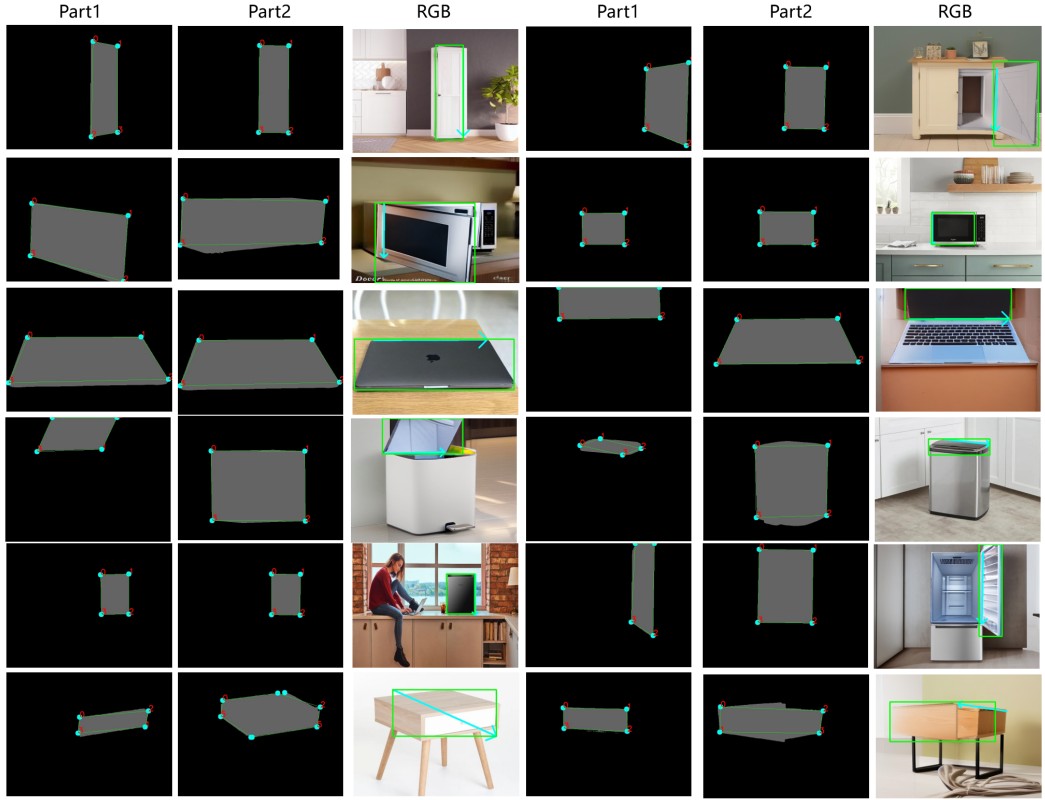

Figure 13: Demonstration of extracting bounding box annotations and rotation/translation axis annotations from part-level masks.

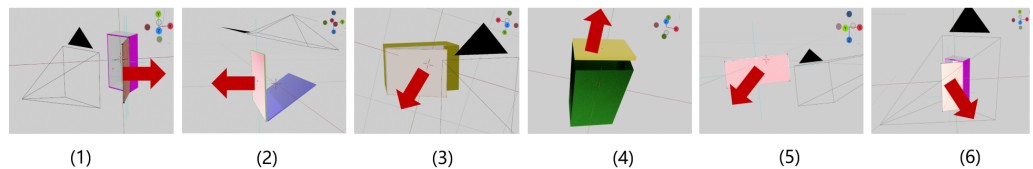

Figure 14: The camera poses and orientations of 6 planes in Blender, (1)-(6) refers to abstract 3D models of cabinet, laptop, microwave, trashcan, drawer, and refrigerator separately. Red arrows are the surface normal directions.

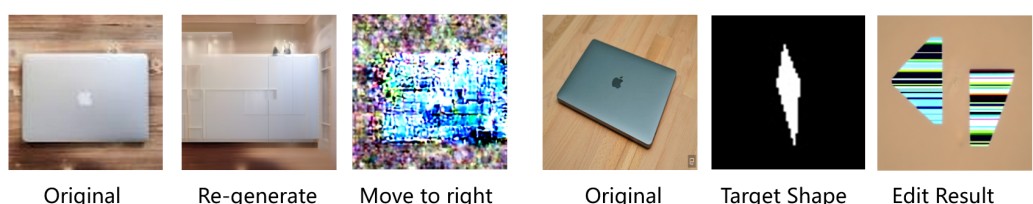

Figure 15: Limitation of the PA-Diffusion model: dealing with low-quality images or large deformation.

