# OpenReview forum: "General Articulated Objects Manipulation in Real Images via Part-Aware Diffusion Process"
_NeurIPS.cc/2024/Conference — NeurIPS 2024 poster_

### Official Review · Reviewer_pkKj · 2024-06-15

**Soundness:** 3
**Presentation:** 3
**Contribution:** 2
**Rating:** 4
**Confidence:** 4

**Summary:**

This paper focuses on manipulating articulated objects in 2D images by leveraging a 2D-3D-2D approach with a diffusion model. Specifically, the authors propose the abstract 3D model to represent the articulated objects and propose dynamic feature maps to transfer seen regions while generating novel areas. The results on image editing and 3D articulated object understanding show the effectiveness of this method.

**Strengths:**

This method is well-motivated: editing images with 3D prior and transferring seen parts is intuitively suitable for articulated object manipulation. The specific techniques applied are also very reasonable and helpful for the task. For example, the design of an abstract 3D model is a good workaround for representing articulated objects instead of using fine-grained 3D meshes. Additional loss functions also make sense.

The experiments are thorough, including two tasks, both qualitative and quantitive, showing the effectiveness of the method in both image editing and down-stream understanding.

The writing is easy to follow, and the appendix provides abundant details.

**Weaknesses:**

My major concern about this approach is its limited application. Specifically, the current process of building abstract 3D models from 2D images might significantly hinder the generalization. The authors create 6 primitive prototypes to represent 6 categories in the dataset. However, this ignores the structure variance among instances in the same category (e.g., a cabinet could have 1-4 drawers and even combine with doors). Therefore, this method actually requires a pre-defined abstract 3D model for each structure. Manually designing an abstract 3D model for each image is clearly impractical, whereas automatically matching 3D prototypes to objects is not trivial, and many works have been done on this topic [1,2].

The process of defining an abstract 3D model for each category also injects crucial priors to the model (i.e., the structure, joint configuration, and kinematic property of the object), which gains advantages compared with baselines that do not require information about the category.

[1] Unsupervised learning for cuboid shape abstraction via joint segmentation from point clouds.

[2] Learning to Infer and Execute 3D Shape Programs.

**Questions:**

1. Could the authors please provide a more detailed description of how to build an abstract 3D model regarding my major concern? I would consider raising or lowering the score based on the way of constructing a 3D model from 2D images.

2. The paper mostly describes and showcases the editing of opening or increasing (e.g., opening door and laptop). I am wondering about the process of 'decreasing' since the design of dynamic feature maps needs to be changed accordingly (e.g., ${M_n}^{Gen}$ should be all zeros, or should it refer to novel backgrounds that are previously occluded by the objects).

3. In line 228, the paper shows that this method could be applied to more objects with more manipulation types. Could the authors provide more description about how this can be done? Do they also require abstract 3D models and manipulation in Blender?

4. Minor: In Figure 4, why do the third column edited images share the same backgrounds while they are not the same as the original one?

**Limitations:**

The author described the limitations. Further discussion is better (Please see the Weakness section).

---

> ### Author Rebuttal · Authors · 2024-08-05
>
> Dear Reviewer,
>
> Thanks for your comments and questions. In the following, we answer all of your questions carefully.
>
> **Q** : Could the authors please provide a more detailed description of how to build an abstract 3D model regarding my major concern? I would consider raising or lowering the score based on the way of constructing a 3D model from 2D images.
>
> **A** : Thank you for your thoughtful question. Generalization is indeed a critical challenge for real-world applications. We believe that the proposed PA-Diffusion model is capable of building Abstract 3D Models for **both simple and complex** articulated objects.
>
> In the paper, we demonstrate that owning to primitive prototypes and Abstract 3D Models, the PA-Diffusion model can handle **a wide range of simple** articulated objects and benefit downstream tasks in robotic applications. The model efficiently incorporates novel shapes, instances, or categories, even when the creation of new prototypes is necessary. Compared to the costly manual collection of robot data, the PA-Diffusion model presents a promising solution for augmenting robot datasets.
>
> Moreover, the PA-Diffusion model can still facilitate the manipulation of **complex objects with extra information**. This research focuses on articulated objects with a single joint, but the PA-Diffusion model can also effectively manage more complex items, such as a cabinet with multiple drawers and doors. This is achieved through an iterative editing process, where each component is modeled and manipulated individually in one step. Sequentially, the results from these manipulations can be concatenated with extra information such as URFD files or descriptions of articulated object structures. Lastly, the PA-Diffusion model edits the images following the same pipeline as simple objects. As depicted in Figure 5, we map both the drawer and cabinet door to a 3D space and manipulate them accordingly. The PA-Diffusion model subsequently edits the images based on these manipulation results.
>
> We appreciate your interest in this aspect of our work, and we hope this clarifies the capabilities of the proposed PA-Diffusion model.
>
> **Q** : The process of defining an abstract 3D model for each category also injects crucial priors to the model (i.e., the structure, joint configuration, and kinematic property of the object), which gains advantages compared with baselines that do not require information about the category.
>
> **A** : While Abstract 3D Models incorporate prior knowledge about the object’s structure, joint configuration, and kinematic properties, we have taken measures to **ensure the fairness of the experimental comparison**.
>
> We selected Imagic, DragDiffusion, MasaCtrl, and Image Sculpting as baselines. Specifically, (1) DragDiffusion requires human interaction, which introduces a more critical prior; (2) MasaCtrl and Image Sculpting utilize the same prior knowledge as our proposed method; (3) Imagic relies solely on text instruction, however, its performance is notably lower compared to the others. Therefore, we believe that this constitutes a fair comparison. We appreciate your question and will include this explanation in the paper.
>
> **Q** : The paper mostly describes and showcases the editing of opening or increasing (e.g., opening door and laptop). I am wondering about the process of ’decreasing’ since the design of dynamic feature maps needs to be changed accordingly (e.g., should be all zeros, or should it refer to novel backgrounds that are previously occluded by the objects).
>
> **A** : For the ’decreasing’ process, our PA-Diffusion model would in-paint backgrounds previously occluded by objects. The situation is similar with moving/rotating objects, as illustrated in Figure 4. When objects are moved or rotated, the resulting blank regions are semantically in-painted based on the surrounding background. Notably, the editing and in-painting are completed in a single step, eliminating the need for an additional in-painting phase.
>
> **Q** : In line 228, the paper shows that this method could be applied to more objects with more manipulation types. Could the authors provide more description about how this can be done? Do they also require abstract 3D models and manipulation in Blender?
>
> **A** : Indeed, for other articulated objects and manipulation tasks, the proposed PA-Diffusion model **adheres to the established pipeline**: constructing abstract 3D models, manipulating them in Blender, and generating edited images. Figure 5, Figure 10, and Figure 11 demonstrate more objects and more manipulation briefly. Please refer to Figure 5, we demonstrate the model’s capability to manipulate non-articulated and non-uniform articulated objects through novel operations, such as breaking cups and opening kitchen pots. Additionally, in Figure 10 and Figure 11, we tested this pipeline on more categories such as doors, toilets, and books.
>
> **Q** : Minor: In Figure 4, why do the third column edited images share the same backgrounds while they are not the same as the original one?
>
> **A** : Thank you. In Figure 4, the variation in backgrounds results from several factors. First, changes in lighting contribute to the variation. Lighting is one of the elements influencing the generation of images with modern Diffusion Models. As we manipulate objects, Diffusion Models account for the altered objects' shapes and adjust the lighting effects accordingly. Second, in-painting the blank regions introduces variations, since in-painting and editing are implemented in the same denoising process. Lastly, the quality of original input images could lead to this as well. To mitigate this, we might enhance the supervision to better preserve the backgrounds during the editing process.
>
> If there exist any other questions please inform us at any time you feel convenient. Thanks a lot.

---

> > ### Comment · Reviewer_pkKj · 2024-08-09
> >
> > Thanks for you well-written reply. It addresses some of my concerns.
> >
> > However, I argue that the construction of the abstract 3D models from 2D images is the most difficult yet important part in your pipeline since 2D-to-3D is definitely not trivial and 3D models contain lots of priors (joint type, joint position, geometry) about the objects for manipulation. From my perspective, having the 3D structure of objects, which is even interactable, usually means the problem is half-solved in robotics. Whereas the approach described in the paper (segmentation module with prototype corner matching) seems too simple to achieve this important and difficult function. Therefore, I still have some concerns about this process.
> >
> > 1. How could the prototype matching solve the ambiguity of 2D information? For example, we know the laptop in the Figure 3 has two parts becasue is a laptop. But why could the prototype matching system identity two planes while only one layer is shown in the image? Does it mean that you give the model the category information? Do the authors  use pre-defined abstract model for laptop instead of matching for each image?
> >
> > 2. I argee with the authors about iterative editing process could probably handle complex objects since the use of diffusion model in this paper is well-designed and reasonable. But I doubt that the 3D abstract model could not be easily obtained for complex objects. Requiring additional URFD files will be a huge drawback of this approach since such data (real image paired with URDF) is rare.
> >
> > 3. Could the authors add an anonymous link to show some results on the prototype matching (image with constrcted 3D abstract model)?
> >
> > Thank you for your rebuttal. Currently I will keep my score due to the concerns above.

---

> ### Author Response · Authors · 2024-08-12
> **Replying to reviewer pkKj**
>
> Dear Reviewer,
>
> Thanks for your questions, we answer them as follows.
>
> **Q** : How could the prototype matching solve the ambiguity of 2D information? For example, we know the laptop in the Figure 3 has two parts becasue is a laptop. But why could the prototype matching system identity two planes while only one layer is shown in the image? Does it mean that you give the model the category information? Do the authors use pre-defined abstract model for laptop instead of matching for each image?
>
> **A** : The 2D information is disentangled and interpreted with common structure knowledge for each object category. **Contrary to the single image to 3D model methods, the PA-Diffusion Model integrates the common structure information for each object category, and then the initial Abstract 3D Models are pre-built in alignment with the information.** For example, the initial Abstract 3D Model of the laptop consists of two planes since a laptop typically consists of a screen and a keyboard, similarly, the initial Abstract 3D Model for a typical microwave includes a body and a door. When editing images, these initial Abstract 3D Models are adjusted to align with each input image. Section 3.2 in the main paper discussed this process in detail. Owing to the straightforward process, the PA-Diffusion Model incorporates novel categories efficiently by extending the structure knowledge and initial Abstract 3D Models.
>
> Yes, the PA-Diffusion Model identifies the object category with the text instructions or by leveraging other large-scale models. Please refer to the link provided, Figure 1 illustrates the process of constructing an Abstract 3D model.  https://drive.google.com/file/d/1W2PRIxt65wxK5NkMqKBOVF2IMgh54XTt/view?usp=sharing
>
> (1)	Using Grounded SAM, we have the semantic segmentation masks of the laptop, Figure 1 (b).
>
> (2)	The initial Abstract 3D Model of the laptop is made up of two 3D planes, one represents the screen and the other represents the body, Figure 1 (c).
>
> (3)	We align the camera view of the Abstract 3D Model and the input image, Figure 1 (d) (e).
>
> (4)	Manipulation can be performed within the 3D space, Figure 1 (f).
>
> **Q** : I argee with the authors about iterative editing process could probably handle complex objects since the use of diffusion model in this paper is well-designed and reasonable. But I doubt that the 3D abstract model could not be easily obtained for complex objects. Requiring additional URFD files will be a huge drawback of this approach since such data (real image paired with URDF) is rare.
>
> **A** : Thanks for your inspective comments. When constructing a 3D model for an entire complex object is necessary, additional spatial geometrical information is required. Utilizing URDF is one of the possible solutions. With advanced large-scale models such as SAM, SAM2, DINO, and so on, there is potential to extract the necessary spatial geometrical information efficiently and economically. In this work, the PA-Diffusion model is specifically designed to support robotic tasks, therefore we mainly focus on everyday articulated objects. Please refer to the attached PDF in the global rebuttal for an example illustrating how the PA-Diffusion model facilitates the generation of sequential robotic manipulation data.
>
> Your question highlights an important aspect that warrants further attention. This is a meaningful point, especially for some downstream applications. We are actively working to incorporate this to further enhance our work.
>
> **Q** : Could the authors add an anonymous link to show some results on the prototype matching (image with constrcted 3D abstract model)?
>
> **A** : Please check the link. Thanks.
>
> https://drive.google.com/file/d/1W2PRIxt65wxK5NkMqKBOVF2IMgh54XTt/view?usp=sharing
>
> If there exists any other questions or confusion, please inform us at any time you feel convenient.

---

> > ### Comment · Reviewer_pkKj · 2024-08-13
> >
> > Thank you for your reply and additional results.
> >
> > Based on the reply, I think the current pipeline is able to solve "typical" objects and requires initial abstract model for each category, which limits the application on diverse objects and more categories.
> >
> > Therefore, I tent to keep my score. I hope the author can improve the 3D abstraction approach in future work. Good luck.

---

### Official Review · Reviewer_YyMw · 2024-07-05

**Soundness:** 3
**Presentation:** 3
**Contribution:** 2
**Rating:** 4
**Confidence:** 4

**Summary:**

This paper presents a pipeline to directly manipulate articulated objects in real images and generate corresponding images in different articulated poses. The proposed method adopts a  2D-3D-2D approach with a heuristic model to obtain 3D information from source images and generate new images based on the diffusion model. No extra fine-tuning and training is required for the proposed method. Quantitative evaluation is presented for this method and qualitative comparison with baseline methods.

**Strengths:**

1. Training-free pipeline for articulated object manipulation is a plus
2. Works out-of-box for real-world objects is a plus
3. The qualitative evaluation demonstrates impressive results

**Weaknesses:**

1. How to obtain 3D information, part-level understanding and structural information is not detailed in Sec. 3.2, which is a key for articulated object manipulation.
2. The text instruction for manipulation generation is based on heuristic models, which may not be flexible.
3. Difficult to follow in Sec. 4.6. Lack of context for the reference work [1][1] (which is [32] in the paper) in this subsection, making it difficult to understand and validate.
   1. As the author mentioned in Sec. 4.1, **NO** training is needed, but Sec. 4.6 talks about dataset spliting and model training, very confusing.
   2. If the "model" is trained in this subsection, what model? with what objectives?
   3. Write the mathematical definition or explain the metrics used in this subsection, Tab. 3.
4. It seems the manipulation is not limited to part-level manipulation, but also applies to the object as a whole, such as move the object to the right. Could this method also extends to object-centric view synthesis? Would be interesting to see experiments on this.
5. It would be better to cite and compare to this similar work [2]

[1] Qian, Shengyi, et al. "Understanding 3d object articulation in internet videos."  *Proceedings of the IEEE/CVF Conference on Computer Vision and Pattern Recognition* . 2022.

[2] Li, Ruining, et al. "Dragapart: Learning a part-level motion prior for articulated objects." *arXiv preprint arXiv:2403.15382* (2024).

**Questions:**

The key questions for this paper focus on the Sec. 3.2.

For abstract 3D model generation. How the proposed method analysis the structural 3D information of the object given images? For example, in the Figure. 3, how does the pipeline obtain the following knowledge from the$Z_T^{init}$:

1. The object is composed of 2 parts, especially the base of the laptop is almost invisible (number of parts discovery)
2. These two parts fit with the plane prototypes (part to prototype fitting)
3. These two planes are connected	(part-level relation discovery)
4. One of the plane can be rotated along the connected axis (joint type estimation between revolute and prismatic)

**Limitations:**

See above

---

> ### Author Rebuttal · Authors · 2024-08-05
>
> Dear Reviewer,
>
> In the following, we answer all your questions in detail. Thanks very much!
>
> **Q** : Difficult to follow in Sec. 4.6. Lack of context for the reference work [1] (which is [32] in the 187 paper) in this subsection, making it difficult to understand and validate. As the author mentioned in Sec. 4.1, NO training is needed, but Sec. 4.6 talks about dataset splitting and model training, very confusing.
>
> **A** : We sincerely apologize for any confusion in our paper and thank you very much for informing us. We will refer to your comments to revise Section 4.6 and clarify this in the final version.
>
> First and foremost, NO training is required for the proposed PA-Diffusion model. Subsection 4.6 serves as an **additional robotic application to illustrate the value of the PA-Diffusion model**. The selected application is the Articulated Object Understanding task, which involves developing a model to estimate the bounding boxes, joint axes, and surface normals of articulated objects [1] in real image/video. We call it the Articulated Object Understanding Model (AOU Model) in the following content.
>
> We generated 3,300 edited images using the proposed PA-Diffusion model, NO training is required in this step. Subsequently, we conducted two separate experiments to validate the quality of the edited images (these edited images are used as training/testing data in the experiments):
>
> We regard the first experiment as a self-test. These 3,300 images were split into training/testing sets to develop/evaluate the AOU Model. The results, presented in Table 2 indicate that more training data can improve the model’s performance, which illustrates that edited images play the same role as real ones
>
> In the second experiment, we select InternetVideo Dataset (a real image dataset) [1] as the baseline. The 3,300 images were mixed with the training set of InternetVideo [1] to train the AOU Model, and evaluate the AOU Model on the testing set (real image as well). The results, shown in Table 3 prove that edited images can enhance the model’s performance, even when evaluated on real images
>
> **Q** : If the "model" is trained in this subsection, what model? with what objectives?
>
> **A** : Thank you for this question. The Articulated Object Understanding Model is trained in subsection 4.6 (not the PA-Diffusion model). The objectives are: (1) Bounding box loss, (2) Axes loss, and (3) Surface Normal loss. Due to the number limitation of character, we will include detailed explanations and definitions in the appendix section.
>
> **Q** : Write the mathematical definition or explain the metrics used in this subsection, Tab. 3.
>
> **A** : The evaluation matrix used in Tab. 3 are: (1) Bounding box evaluation matrix, (2) Axes evaluation matrix - Euclidean distance and Angular distance Score, and (3) Surface normal evaluation matrix. Thresholds are set for these metrics, and we utilize the accuracy to evaluate the model. Due to the number limitation of characters, we will include detailed explanations and definitions in the appendix section.
>
> **Q** : It seems the manipulation is not limited to part-level manipulation, but also applies to the object as a whole, such as move the object to the right. Could this method also extends to object-centric view synthesis? Would be interesting to see experiments on this.
>
> **A** : Thank you for this insightful suggestion. **Given that objects can be manipulated in various ways, the proposed method is intended to support object-centric view synthesis**. We are actively working on implementing this feature, as it represents a compelling aspect of our research.
>
> **Q** : It would be better to cite and compare to this similar work [2]
>
> **A** : Thank you. We will cite [2] and include an analysis of their work in our paper. Reference [2] proposed an innovative method for manipulating articulated objects by modifying feature/attention maps to perform various editing tasks. In comparison, our approach maps the objects to 3D space for manipulation and then modifies the initial inverted noise maps to (1) preserve the objects’ appearance, and (2) generate novel-appearing parts.
>
> **Q** : The key questions for this paper focus on the Sec. 3.2. For abstract 3D model generation. How the proposed method analysis the structural 3D information of the object given images? For example, in the Figure. 3, how does the pipeline obtain the following knowledge from the : The object is composed of 2 parts, especially the base of the laptop is almost invisible (number of parts discovery)
>
> These two parts fit with the plane prototypes (part to prototype fitting)
>
> These two planes are connected (part-level relation discovery)
>
> One of the plane can be rotated along the connected axis (joint type estimation between revolute and prismatic)
>
> **A** : Thank you for your insightful question. The structural information is pre-defined for each articulated object category according to the common knowledge. For instance, a laptop consists of two parts and a rotation axis. Based on this knowledge, objects in real images are segmented using Grounded SAM, and then abstract 3D models are created referring to the part-level segmentation masks. This procedure ensures that the structural knowledge is incorporated into the abstract 3D models. The flexibility of the proposed PA-Diffusion model further allows for the efficient incorporation of novel structural information and objects.
>
> If there exist any other questions or confusion, please inform us at any time you feel convenient. Your questions and suggestions are so beneficial to our research. Thank you very much.
>
> Reference
>
> [1] Qian, Shengyi, et al. "Understanding 3d object articulation in internet videos." Proceedings of the IEEE/CVF Conference on Computer Vision and Pattern Recognition, 2022
>
> [2] Li, Ruining, et al. "Dragapart: Learning a part-level motion prior for articulated objects." arXiv preprint arXiv:2403.15382, 2024

---

> ### Comment · Reviewer_YyMw · 2024-08-10
>
> Thank you for your detailed reply. It address some of my concern.
>
> However, I still have some questions given the reply on the 3D modelling from image. If I understand the reply correctly, these 4 steps I asked in the question is done heuristically.
>
> ```
> 1. The object is composed of 2 parts, especially the base of the laptop is almost invisible (number of parts discovery)
> 2. These two parts fit with the plane prototypes (part to prototype fitting)
> 3. These two planes are connected (part-level relation discovery)
> 4. One of the plane can be rotated along the connected axis (joint type estimation between revolute and prismatic)
> ```
> And the "modelling" task done in this paper is part-level segmentation and pose fitting.
>
> If that's the case, I would agree with the reviewer pkKj. A 2D to 3D understanding for articulated object is the key for manipulation. I would suggest the author to reframe a bit on the problem they are targeting, perhaps try to avoid the "3D modelling" task and focus more on image synthesis in the pipeline and writing.
>
> I will keep my rating based on the above concern.

---

> ### Author Response · Authors · 2024-08-12
> **replying to reviewer YyMw**
>
> Dear Reviewer,
>
> Thank you for your questions and comments. Please check the following answers.
>
> **Q** : However, I still have some problems given the reply on the 3D modelling from image. If I understand the reply correctly, these 4 steps I asked in the question is done heuristically.
>
> 1. The object is composed of 2 parts, especially the base of the laptop is almost invisible (number of parts discovery)
>
> 2. These two parts fit with the plane prototypes (part to prototype fitting)
>
> 3. These two planes are connected (part-level relation discovery)
>
> 4. One of the plane can be rotated along the connected axis (joint type estimation between revolute and prismatic)
> And the "modelling" task done in this paper is part-level segmentation and pose fitting.
>
> If that's the case, I would agree with the reviewer pkKj. A 2D to 3D understanding for articulated object is the key for manipulation. I would suggest the author to reframe a bit on the problem they are targeting, perhaps try to avoid the "3D modelling" task and focus more on image synthesis in the pipeline and writing.
>
> **A** : Your understanding is accurate. The structural information for commonly encountered object categories is pre-collected based on established knowledge. Following the four steps you mentioned, Abstract 3D Models integrate this information to facilitate manipulation and editing tasks. Due to the simple process, the PA-Diffusion model can be extended to incorporate additional structural information and novel objects with minimal complexity.
>
> The transition from 2D to 3D modeling is one of the primary contributions of this work, which benefits both the image editing task and the downstream tasks in robot scenarios. Compared to other state-of-the-art editing methods, the PA-Diffusion model demonstrates superior capability in object editing, largely due to the support provided by Abstract 3D Models.
>
> Additionally, utilizing the Abstract 3D Model offers several advantages including improved accuracy, efficiency, ease of manipulation, the ability to handle multiple categories, and incorporating novel categories quickly.
> Beyond image editing, these advantages suggest that the PA-Diffusion Model holds promise in reducing the reliance on expensive robotic manipulation data. Please check the attached PDF file in the global rebuttal, there is an example of how to generate sequence robot manipulation data with the PA-Diffusion Model.
>
> We appreciate your suggestion and will take your advice to focus more on image synthesis in the paper.

---

> > ### Comment · Reviewer_YyMw · 2024-08-13
> >
> > Thank you for your reply. I would keep my rating based on the discussion.

---

### Official Review · Reviewer_N2Nu · 2024-07-06

**Soundness:** 3
**Presentation:** 3
**Contribution:** 4
**Rating:** 7
**Confidence:** 3

**Summary:**

This paper introduces a novel diffusion-based method for manipulating articulated objects in real-world images from input text guidance or human interaction. There are three main contributions in this paper: (i) the authors introduce the concept of an Abstract 3D Model, which eliminates the requirement for a 3D model by using primitives such as cuboids, boxes, etc to represent various types of articulated objects; (ii) the authors present dynamic feature maps, which transfer the seen parts from the input image to the generated image and only generate the novel parts, which is very intuitive; (iii) the test benchmark includes 660 object samples and a total of 3300 images.

**Strengths:**

- The most significant strength is the training-free method, i.e, given an input image at test time, the proposed method uses texture and style guidance loss to optimize the output image using only different pre-trained models. This training-free approach allows easy generalization to novel categories and different types of objects.

- I like the concept of the Abstract 3D Model, as acquiring 3D models of in-the-wild objects can be challenging and costly (although I have some doubts, as noted in the Weaknesses section below).

- The proposed dynamic feature maps are also very intuitive, ensuring that the generated image remains consistent with the seen parts of the input image while generating only the novel parts.

- The proposed methods outperform previous works on several metrics and benchmarks.

**Weaknesses:**

-  While I like the concept of the Abstract 3D Model, I am not sure about its effectiveness when the input objects have shapes that do not conform to the available primitives, such as a round table, etc. In such cases, the part-level masks and sketch maps can be very different.

- The authors' observation (L218) and the qualitative results do not well explain why articulated objects like refrigerators or furniture always appear empty when opened. I would expect it is randomly generated, with some cases showing objects, fruits and with other cases empty.

- The generated shadows in the output images also seem under-analyzed. In some instances, the shadows do not change even though the object has been articulated (e.g., some furniture samples in Figure 6).

**Questions:**

Overall, I found the paper is well-written, and the contributions are strong and clear. However, I have a few questions to better understanding of the proposed methods and address the weaknesses mentioned above:

- Can the proposed methods effectively handle input objects whose shapes do not match the available primitives?

- Why are the drawers of opened objects consistently empty?

- Could you provide an analysis of shadows in the generated images, as this is a crucial factor in distinguishing whether the images are generated or real?

**Limitations:**

Yes, the authors clearly stated the limitation of proposed method (i) when input images are blurry, in low-resolution, (ii) when input objects are deformable and or fluids.

---

> ### Author Rebuttal · Authors · 2024-08-05
>
> Dear Reviewer,
>
> Thank you for all of your kind suggestions and questions. In the following, we answer all of your questions in detail.
>
> **Q** : Can the proposed methods effectively handle input objects whose shapes do not match the available primitives?
>
> **A** : Thank you for your valuable question. Generalization is indeed a critical factor, and the proposed PA-Diffusion model is designed to handle various articulated objects from the following three aspects:
>
> First, **Primitive Prototype Library** encompasses many categories of common articulated objects by utilizing prototypes. As illustrated in Figure 4, Figure 10 and Figure 11, six primitive prototypes can adequately represent objects such as laptops, doors, and other categories. By extending the library, we can accommodate additional categories and instances, thereby significantly enhancing support for robotic tasks.
>
> Second, when the shapes of objects **mismatch significantly** from existing primitive prototypes, we can create novel prototypes as needed, which is both efficient and straightforward.
>
> Lastly, leveraging current **image-to-3D methods** gives us another viable option. As demonstrated in Figure 5, we utilize ZERO123 [1] to generate 3D models for non-uniform objects before proceeding with image editing, and the edited image is still reasonable and high-fidelity.
>
> **Q** : Why are the drawers of opened objects consistently empty?
>
> **A** : Thank you. We consider this consistency an **advantage** of the proposed PA-Diffusion model. As illustrated in Figure 4, when objects are manipulated continuously, the style and appearance of the novel-appearing parts remain consistent. This consistency makes the method suitable for video generation.
>
> This consistency is achieved through **the text prompts** and **sketch conditions** used during the image editing process. For instance, when manipulating drawers, the text prompt "a photo of an opened drawer" is applied uniformly across all test cases, while the sketch map consistently depicts an empty drawer. These two factors contribute to the uniform appearance of opened drawers.
>
> **Q** : Could you provide an analysis of shadows in the generated images, as this is a crucial factor in distinguishing whether the images are generated or real?
>
> **A** : This is an inspective question. Our primary focus is on manipulating articulated objects in images, meanwhile preserving the appearance of the background. The shadow region is considered part of the background and is therefore maintained during the image editing process. This challenge could be addressed by incorporating additional prompts or other shadow manipulation techniques [2], [3].
>
> We appreciate your suggestion in guiding our research, and in the next steps, we will consider the impact of shadows to enhance the quality of the edited images, which is indeed an important aspect.
>
> Reference
>
> [1] Ruoshi Liu, et. al, Zero-1-to-3: Zero-shot One Image to 3D Object, ICCV, 2023
>
> [2] Qingyang Liu, et. al, Shadow Generation for Composite Image Using Diffusion Model, https://arxiv.org/pdf/2403.15234
>
> [3] Jinting Luo, et. al, Diff-Shadow: Global-guided Diffusion Model for Shadow Removal, https://www.arxiv.org/pdf/2407.16214

---

### Official Review · Reviewer_1MgD · 2024-07-08

**Soundness:** 3
**Presentation:** 3
**Contribution:** 3
**Rating:** 6
**Confidence:** 5

**Summary:**

The paper proposes a method for accurately generating edited images for manipulating articulated objects, effectively avoiding hallucinations. The overall idea and approach are very interesting. The implementation of the method is particularly impressive, presenting a novel articulated object manipulation technique that covers common object categories and supports arbitrary manipulation.

**Strengths:**

The idea is interesting

**Weaknesses:**

please refer to question section

**Questions:**

1. The description of section discussing manipulation in L122 is unclear. For instance, if an object has many movable parts, how is it determined which part should be moved? Taking the example in the figure, a laptop has two planes. If the text says "open laptop," how is it determined which plane should be moved?
2. The goal of "Abstract 3D Model" in L105 should be made clearer. For example, I am wondering why single-image reconstruction methods are not used to obtain the 3D model. Although I understand that using prototype reconstruction methods facilitates subsequent manipulation of arbitrary parts, I believe the authors should clearly explain the reasons and purposes behind each method detail.
3. In the abstract and introduction, the authors repeatedly mention that edited images are significant for robotic manipulation learning. Could authors explain in detail why edited images are beneficial for robotic manipulation? If conducting experiments is challenging, analyzing the advantages of image editing for robotics is also acceptable. For example, what limitations do existing image goal-conditioned manipulation policies have, and can this editing method address these limitations?
4. In Figure3, there are masks M in both top and bottom row. But why do the image M^{gen} on the top and image M_{s}^{gen} on the bottom look so different? I suppose mask should be in binary mask, but why M^{gen}  is so colorful?

**Limitations:**

Yes

---

> ### Author Rebuttal · Authors · 2024-08-05
>
> Dear Reviewer,
>
> Thanks very much for your valuable suggestions. In the following, we answer all of your questions in detail and carefully.
>
> **Q** : The description of section discussing manipulation in L122 is unclear. For instance, if an object has many movable parts, how is it determined which part should be moved? Taking the example in the figure, a laptop has two planes. If the text says "open laptop," how is it determined which plane should be moved?
>
> **A** : Thank you for your insightful question, we will incorporate more explanation into the paper for clarity. In our work, we pre-define the movable parts in **the mapping table**, aligning with **common manipulations encountered in daily life**. As illustrated in Figure 9 in the appendix session, the mapping table specifies which parts should be moved based on the provided text instructions. For instance, the instruction ‘open laptop’ pertains to manipulating the screen plane of the laptop.
>
> Furthermore, the PA-Diffusion model is capable of handling multiple movable parts as well. As demonstrated in Figure 5, the text instruction ‘open drawer and open cabinet’ allows for the concurrent opening of both the drawer and the cabinet. More generally, the mapping table can be extended to manage multiple movable parts within a single object, thereby enhancing the model’s flexibility and applicability. Human interaction is also supported.
>
> **Q** : The goal of "Abstract 3D Model" in L105 should be made clearer. For example, I am wondering why single-image reconstruction methods are not used to obtain the 3D model. Although I understand that using prototype reconstruction methods facilitates subsequent manipulation of arbitrary parts, I believe the authors should clearly explain the reasons and purposes behind each method in detail.
>
> **A** : Thanks for the instructive suggestion. In this work, we highlight several advantages of using Abstract 3D Models compared to single-image reconstruction methods:
>
> 1) Abstract 3D Models demonstrate **better accuracy** in generating 3D models across various object categories.
>
> 2) Abstract 3D Models are **easier** to manipulate, particularly for articulated objects.
>
> 3) Abstract 3D Models provide a clear definition of object parts.
>
> 4) Abstract 3D Models allow efficient coverage of **novel** instances and categories, whereas single-image reconstruction methods require fine-tuning with additional samples.
>
> 5) Using Abstract 3D Models makes the image editing process **time-efficient** enough to support other downstream tasks, which is previously challenging.
>
> Please refer to Columns 3 and 4 in Figure 12, the reconstructed 3D models produced by ZERO123 [6] are inadequate for supporting image editing. Besides, manual effort is required to split and manipulate the 3D mesh model, which is tedious and costly. For further comparison and analysis, please refer to subsection A4 in the appendix section.
>
> Nonetheless, we recognize the value of single-image reconstruction methods as a complement to Abstract 3D Models in certain situations. As shown in Figure 5, we selected ZERO123 [6] to generate the 3D model for the toy shark with a non-uniform shape. We observe that the PA-Diffusion can create high-fidelity edited images with this 3D model as well.
>
> **Q** : In the abstract and introduction, the authors repeatedly mention that edited images are significant for robotic manipulation learning. Could authors explain in detail why edited images are beneficial for robotic manipulation? If conducting experiments is challenging, analyzing the advantages of image editing for robotics is also acceptable. For example, what limitations do existing image goal-conditioned manipulation policies have, and can this editing method address these limitations?
>
> **A** : Thanks. We respectfully explain how the PA-Diffusion model can benefit robotic manipulation tasks in two key areas: **sub-goal generation** and **data augmentation**. **Please check the attached file in the global rebuttal, there is a robot manipulation demo**.
>
> Regarding sub-goal generation, recent works [1], [2], [3] have introduced the concept of generating sub-goal conditions to assist with long-horizon tasks. This approach improves the overall manipulation success rate.
>
> For data augmentation, [4] proposed augmenting datasets by changing the appearance of objects or backgrounds. In comparison, the PA-Diffusion model supports arbitrary articulated object manipulation in real images, **significantly enriching** the dataset and supporting more complex robotic tasks.
>
> Moreover, directly predicting the manipulation process and extracting manipulation policies have emerged as promising methods to improve the success rate [5]. The PA-Diffusion model shows great potential in enhancing these methods by generating high-quality manipulation processes for various objects. We are currently conducting further research on robotic manipulation alignment with this approach.
>
> **Q** : In Figure3, there are masks M in both top and bottom row. But why do the image Mgen on the top and image Mgen on the bottom look so different? I suppose mask should be in binary mask, but why Mgen 118 is so colorful?
>
> **A** : In Figure 3, Bottom Seen mask refers to top Part1 mask, bottom Novel mask refers to top (Part2 - Part1) mask. We have made the masks colorful to distinguish the seen and novel-appearing parts, aiming to simplify and clarify the editing process.
>
> We appreciate all of these questions and will incorporate these analyses into our paper.
>
> Refs are shrunk.
>
> [1] Subgoal Diffuser: Coarse-to-fine Subgoal Generation to Guide Model Predictive Control for Robot Manipulation
>
> [2] Zero-Shot Robotic Manipulation with Pretrained Image-Editing Diffusion Models
>
> [3] OpenVLA: An Open-Source Vision-Language-Action Model
>
> [4] GenAug: Retargeting behaviors to unseen situations via Generative Augmentation
>
> [5] Generative Image as Action Models
>
> [6] Zero-1-to-3: Zero-shot One Image to 3D Object

---

> > ### Comment · Reviewer_1MgD · 2024-08-10
> > **Tend to accept**
> >
> > The author's response addresses my concerns comprehensively. I believe this is a strong paper that merits acceptance at NeurIPS. Good luck!

---

### Author Rebuttal · Authors · 2024-08-05

Dear Reviewers,

We sincerely appreciate all of your insightful and valuable questions and suggestions. We have made a concerted effort to address each query thoroughly and have revised the paper comprehensively following your recommendations. Your perceptive insights have undeniably enriched our work.

It is widely acknowledged that modern deep learning-based robotic manipulation models can perform a variety of tasks in real-world environments [1-5]. However, the available datasets are still insufficient for developing a truly generalizable model since collecting large-scale robotic datasets is expensive. In light of this limitation, the proposed PA-Diffusion model is introduced to manipulate various articulated objects in real images, which approaches this challenge from **a novel perspective by utilizing low-cost image editing techniques to mitigate the reliance on expensive robotic data**.

In this work, we demonstrate how the PA-Diffusion model manipulates various articulated objects in real images, and the edited images can serve as valuable resources for other computer vision and robotics-related tasks. Furthermore, the flexibility of the PA-Diffusion model allows for easy adaptation to new objects, positioning it as a promising approach to bridge the gap between general robotic manipulation and the challenges posed by limited data availability.

To illustrate this point, we present an intuitive example in the field of robot data generation. As demonstrated in Figure 1 in the attached file, the PA-Diffusion model effectively manipulates the microwave to key states as represented in (a), and then the 3D end-effector pose can be extracted in each state (b). Consequently, within robot simulators such as RLBench, the robot arm, and end-effector are moved to specific poses (c). During this process, the necessary information, including joint positions, point clouds, and so on, can be loaded and recorded by reading the current state from the simulator. In the end, a complete manipulation sample is generated from a single RGB image.

When compared with relying on simulators [6],[7],[8],[9] to synthesize manipulation samples, the PA-Diffusion model offers **more versatile samples featuring different objects and backgrounds**. This ongoing work holds significant potential and merits further investigation.

We are truly grateful for your invaluable guidance. Thank you again.

Reference

[1] M. Shridhar, et. al, Perceiver-actor: A multi-task transformer for robotic manipulation. In Conference on Robot Learning, pages 785–799. PMLR, 2023

[2] C. Chi, et. al, Diffusion policy: Visuomotor policy learning via action diffusion. arXiv preprint arXiv:2303.04137, 2023.

[3] Z. Xian, et. al, Chaineddiffuser: Unifying trajectory diffusion and keypose prediction for robotic manipulation. In 7th Annual Conference on Robot Learning, 2023.

[4] X. Ma, et. al, Hierarchical diffusion policy for kinematics aware multi-task robotic manipulation. arXiv preprint arXiv:2403.03890, 2024.

[5] V. Vosylius, et. al, Render and diffuse: Aligning image and action spaces for diffusion-based behaviour cloning, 2024.

[6] E. Rohmer, et. al, Coppeliasim (formerly v-rep): a versatile and scalable robot simulation framework. In Proc. of The International Conference on Intelligent Robots and Systems (IROS), 2013. www.coppeliarobotics.com

[7] S. James, et. al, Pyrep: Bringing v-rep to deep robot learning. arXiv preprint arXiv:1906.11176, 2019

[8] S. James, et. al, Rlbench: The robot learning benchmark learning environment. IEEE Robotics and Automation Letters, 5(2):3019–3026, 2020

[9] Soroush Nasiriany, et. al, RoboCasa: Large-Scale Simulation of Everyday Tasks for Generalist Robots, Robotics: Science and Systems, 2024

---

### Decision · Program_Chairs · 2024-09-25

**Decision:**

Accept (poster)

**Comment:**

This manuscript describes an approach to image generation for articulated objects. The approach proceeds by providing guidance to the image interpretation process in the form of an abstract 3D model which captures the form of the articulation and allows the system to generate results that capture the intended variation. The paper was reviewed by a panel of experts who felt that the method was well presented and they appreciated the fact that the method was able to produce reasonably compelling results with relatively little user input. That being said there was significant discussion about the generality of the method and the need for the user to provide high level guidance. It is not clear from the results whether the system could be generalized to handle objects with larger numbers of degrees of freedom. While robotics is mentioned as a possible application area the experimental results do not show any examples of application to robotic arms with 3 or more joints. Nonetheless, there was a consensus that the method appeared to work reasonably well within the limited domain of abstract 3D models that were presented and readers may appreciate the insights that it offers